# Metabolic crosstalk between skeletal muscle cells and liver through IRF4-FSTL1 in nonalcoholic steatohepatitis

Shanshan Guo[1,8], Yonghao Feng [1,8], Xiaopeng Zhu[2,8], Xinyi Zhang[3,8], Hui Wang[4], Ruwen Wang[5], Qiongyue Zhang[6], Yiming Li[6], Yan Ren [7], Xin Gao[2], Hua Bian[2] ✉, Tiemin Liu[1,2,3,4] ✉, Huanqing Gao[1] ✉ & Xingxing Kong [1,4] ✉

Inter-organ crosstalk has gained increasing attention in recent times; however, the underlying mechanisms remain unclear. In this study, we elucidate an endocrine pathway that is regulated by skeletal muscle interferon regulatory factor (IRF) 4, which manipulates liver pathology. Skeletal muscle specific IRF4 knockout (F4MKO) mice exhibited ameliorated hepatic steatosis, inflammation, and fibrosis, without changes in body weight, when put on a nonalcoholic steatohepatitis (NASH) diet. Proteomics analysis results suggested that follistatin-like protein 1 (FSTL1) may constitute a link between muscles and the liver. Dual luciferase assays showed that IRF4 can transcriptionally regulate FSTL1. Further, inducing FSTL1 expression in the muscles of F4MKO mice is sufficient to restore liver pathology. In addition, co-culture experiments confirmed that FSTL1 plays a distinct role in various liver cell types via different receptors. Finally, we observed that the serum FSTL1 level is positively correlated with NASH progression in humans. These data indicate a signaling pathway involving IRF4-FSTL1-DIP2A/CD14, that links skeletal muscle cells to the liver in the pathogenesis of NASH.

Nonalcoholic steatohepatitis (NASH) is characterized by liver steatosis, inflammation, hepatocyte ballooning injury, and varying degrees of fibrosis[1–3]. NASH is associated with a higher risk of progression to cirrhosis and hepatocellular carcinoma and is associated with metabolic diseases, including type 2 diabetes and metabolic syndrome[4,5]. Increased physical conditioning has been observed to potentially improve hepatic metabolism independent of an individual's body weight[6]. Apart from adipose tissue function and other targets of insulin activity, skeletal muscle physiology is closely integrated with overall energy homeostasis and calorie disposal. Skeletal muscle, acting as an endocrine organ, can secrete various myokines that affect other metabolic tissues, including liver tissue[7]. While some myokines (e.g., IL-6, myostatin, and activin) have been found to play roles in liver steatosis, fibrosis, and cancer[8–12], many others are yet to be comprehensively investigated.

Interferon regulatory factor (IRF) 4, a member of the IRF family, is known for its role in the regulation of immune cell development and function[13,14]. It is also a transcriptional regulator of lipolytic genes in adipose tissue[15,16], and partners with the co-activator PGC-1α in regulating thermogenesis in brown adipose tissue (BAT)[17,18]. Recently, we

[1]Department of Endocrinology and Metabolism, State Key Laboratory of Genetic Engineering, School of Life Sciences, Huashan Hospital, Fudan University, Shanghai 200040, China. [2]Department of Endocrinology and Metabolism, Zhongshan Hospital, Fudan University, Shanghai 200032, China. [3]Human Phenome Institute, Fudan University, Shanghai 201203, China. [4]Shanghai Key Laboratory of Metabolic Remodeling and Health, Institute of Metabolism & Integrative Biology, Fudan University, Shanghai 200438, China. [5]School of Kinesiology, Shanghai University of Sport, Shanghai 200438, China. [6]Department of Endocrinology and Metabolism, Huashan Hospital, Fudan University, Shanghai 200040, China. [7]Experiment Center for Science and Technology, Shanghai University of Traditional Chinese Medicine, Shanghai 201203, China. [8]These authors contributed equally: Shanshan Guo, Yonghao Feng, Xiaopeng Zhu, Xinyi Zhang. ✉e-mail: zhongshan_bh@126.com; tiemin_liu@fudan.edu.cn; gao_hq@fudan.edu.cn; kongxingxing@fudan.edu.cn

found that skeletal muscle-specific IRF4 knockout (F4MKO) mice fed a chow diet exhibited normal body weight and insulin sensitivity, but increased exercise capacity[19]. However, when put on a high-fat diet (HFD), F4MKO mice appeared to be protected from diet-induced obesity, and showed improved glucose tolerance and increased insulin sensitivity[20]. Further investigations are needed to determine whether the ablation of IRF4 in skeletal muscle would affect liver metabolism, given that insulin resistance in skeletal muscles has been linked to increased hepatic de novo lipogenesis and hepatic steatosis in the elderly[21].

Follistatin-like protein 1 (FSTL1), a secretory glycoprotein, is a potential target for steatosis-associated liver fibrosis[22]. Macrophage FSTL1 has been shown to promote liver fibrosis by inducing M1 polarization and inflammation[23]. The overexpression of Fstl1 in the liver could promote the expression of proinflammatory cytokines TNF-α, IL-1β, and IL-6[24]. Silencing Fstl1 using shRNA can reduce Col1a1 mRNA expression and macrophage accumulation in carbon tetrachloride injury-mediated liver fibrosis in mice[25]. Meanwhile, FSTL1 has been identified as a muscle-derived secretory myokine[7,26], and skeletal muscle is a major source of circulating FSTL1[27]. Dynamic resistance exercise was found to increase the expression of skeletal muscle-derived FSTL1, which can supplement the insufficiency of cardiac FSTL1 expression, induce cardiac angiogenesis, and promote cardiac rehabilitation in myocardial infarction rats[26]. Although FSTL1 has various functions, it remains unclear which receptors are required for FSTL1-mediated inflammatory responses, lipid metabolism, and fibrosis in the liver.

Inter-organ crosstalk has gained increasing attention recently. In this study, we proposed a signaling pathway that potentially regulates the communication between skeletal muscle and the liver, especially in NASH models. First, we found that IRF4 expression was significantly upregulated in the skeletal muscle of NASH mice. Second, F4MKO mice showed ameliorated hepatic pathology when administered a NASH diet. Third, proteomics analysis and dual luciferase assays results indicated that FSTL1 acts as a myokine that mediates this crosstalk. Moreover, findings from cell co-culture experiments suggested that DIP2A and CD14 could be receptors of FSTL1 in the liver to mediate its function. Lastly, the serum FSTL1 level was increased in patients with NASH compared with that in healthy individuals. Taken together, these results revealed a signaling pathway from skeletal muscles to the liver via the IRF4-FSTL1-DIP2A/CD14 pathway in the pathogenesis of NASH.

## Results

### Ablation of IRF4 expression in skeletal muscles ameliorated NASH

To confirm whether IRF4 expression was altered in the skeletal muscles of the NASH model, we measured the mRNA and protein expression levels in skeletal muscle and liver tissues from high-glucose and high-fat-diet-fed NASH mouse models. IRF4 protein and mRNA expression was increased in the gastrocnemius (GAS) (Fig. 1a, b), without significant changes in the liver of NASH mice (Supplementary Fig. 1a, b). As exercise can attenuate the NASH phenotype (Supplementary Fig. 1c–e), IRF4 expression was found to be decreased in GAS in NASH mice subjected to exercise (Fig. 1c, d). These results indicated that skeletal muscle IRF4 expression was associated with the hepatic pathology of NASH.

To investigate whether the ablation of IRF4 in skeletal muscles ameliorated NASH progression, we generated skeletal muscle-specific IRF4 knockout (F4MKO) mice, as previously described[19]. Male mice were then fed a Western diet and high-fructose diet for 24 weeks to induce NASH, as reported before[28]. No differences in body weight and liver weight were observed between F4MKO and Flox mice (Supplementary Fig. 1f–h) fed the NASH diet. Liver tissue morphology and H&E staining results showed a mild difference in hepatic steatosis (Fig. 1e, f,

Supplementary Fig. 1i). Moreover, unlike HFD-fed mice[20], the F4MKO mice showed no changes in fat and muscle depots compared with Flox mice (Supplementary Fig. 1j, k). Similarly, no changes were observed in oxygen consumption, carbon dioxide production, respiratory exchange ratio (RER), and energy expenditure (EE) (Supplementary Fig. 1l–o). GAS fiber type-related genes and H&E staining also did not reveal any changes in the muscle fiber type or muscle cross-sectional area (Supplementary Fig. 1p, q). This was further illustrated by the NASH/NAFLD activity score (NAS score) in the liver of Flox and F4MKO mice (Fig. 1f). In addition, we observed that F4MKO mice showed lower serum ALT and AST levels (Fig. 1g, h), which were indicators of liver damage. Collectively, these data suggested that deletion of IRF4 in skeletal muscles could improve NASH progression. The amelioration of hepatic steatosis and liver damage was also observed in female mice (Supplementary Fig. 1r–t).

### F4MKO mice showed a reduction in lipid accumulation, lower inflammation levels, and decreased fibrosis when fed the NASH diet

To further investigate the potential hepatic pathologies, we assessed lipid metabolism, inflammation, macrophage infiltration, and fibrosis in F4MKO and control mice. As anticipated, the levels of triglycerides and cholesterol were lower in the liver of F4MKO mice (Fig. 2a, Supplementary Fig. 2a, b), whereas the level of plasma triglycerides and cholesterol did not change (Supplementary Fig. 2c, d). Concomitant with this, lipogenic genes (Srebp1c, Scd1, Nr1h3, Fas) were downregulated in the liver of F4MKO mice compared with that in control mice, whereas fatty acid oxidation genes (Pparα, Cpt1a) were upregulated (Fig. 2b). To address macrophage infiltration in the liver, we measured F4/80+ Kupffer cells/macrophages in the liver. F4/80+ staining showed that there were fewer F4/80+-positive cells in F4MKO mice than in control mice (Fig. 2c). In addition, the M1 macrophage maker gene Nos2 was decreased and M2 marker genes Clec10a was upregulated. In line with this, the inflammatory genes Il-6, Il-1β, and Tnf were downregulated and anti-inflammatory gene Il-10 was upregulated in samples obtained from F4MKO mice (Fig. 2d). Furthermore, liver fibrosis was suppressed in the samples of F4MKO mice assessed by Sirius Red staining, along with the downregulation of key genes of fibrosis (Acat2, Col1a1, Col1a2, Timp2, and Tgfβ) (Fig. 2e, f).

Next, we assessed whether skeletal muscle tissues from F4MKO mice could directly regulate liver function by using a co-culture system (Supplementary Fig. 2e). We first generated cells to mimic NASH in vivo by treating hepatocytes and macrophages with palmitic acid (PA), and treating HSC-T6 cells with TGFβ (Supplementary Fig. 2f–i). When primary hepatocytes were co-cultured with GAS from F4MKO or control mice, lipid droplet accumulation was found to decrease (Fig. 2g), and the cellular TG level also decreased significantly in the F4MKO group relative to that in the control group (Fig. 2h). This phenomenon was also observed in HepG2 and AML12 cells (Supplementary Fig. 2j–m). In addition, the expression of inflammation-related genes in macrophages and fibrosis in HSC-T6 cells decreased when the cells were co-cultured with GAS from F4MKO mice (Fig. 2i, j). These data indicated that IRF4-specific knockout in skeletal muscles led to a reduction in lipid accumulation, inflammation, and fibrosis in liver tissues and in cells in vitro.

### The expression of FSTL1, a myokine, was decreased in F4MKO mice and transcriptionally regulated by IRF4

The co-culture system indicated that skeletal muscles can directly communicate with liver tissues. We performed serum proteomics using samples from F4MKO and control mice and identified 376 differentially expressed proteins corresponding to a fold change >1.5 or fold change <0.67. This included 194 upregulated proteins and 182 downregulated proteins (Supplementary Fig. 3a). We then compared genes encoding these proteins with those differently

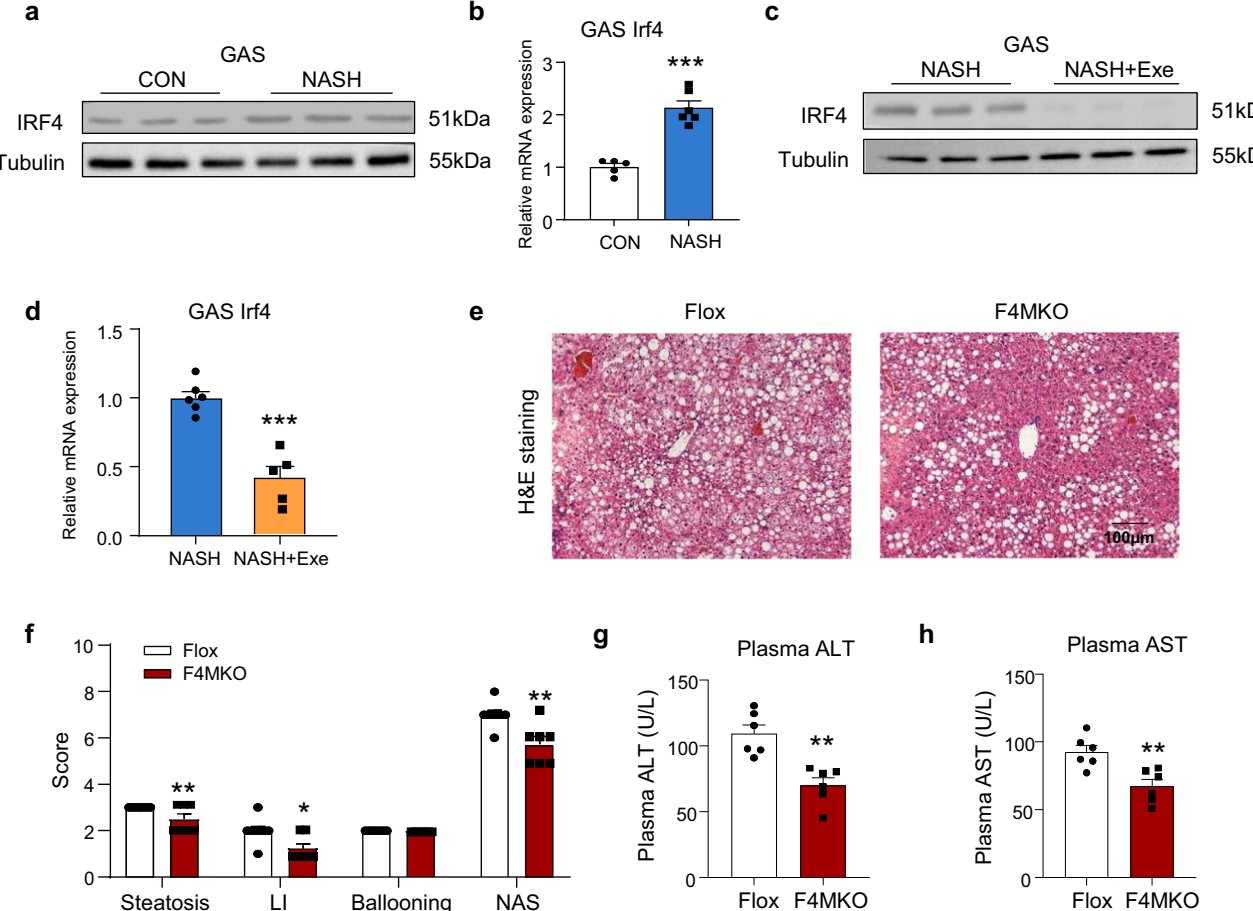

**Fig. 1 | Ablation of IRF4 in skeletal muscle attenuates NASH. a**, **b** Western blot ($n = 3$) and QPCR (CON, $n = 5$ biologically independent animals; NASH, $n = 6$ biologically independent animals) analysis of the expression of IRF4 in skeletal muscle of control (CON) and NASH mice. **c**, **d** Western blot ($n = 3$) and QPCR (NASH, $n = 6$ biologically independent animals; NASH+Exe, $n = 5$ biologically independent animals) analysis of the expression of IRF4 in skeletal muscle of NASH and NASH+ Exercise (NASH+Exe) mice. **e** H&E staining of liver in Flox and F4MKO mice on NASH diet (Flox, $n = 8$ biologically independent animals; MKO, $n = 7$ biologically independent animals. Scale bars, 100 μm). **f** Quantification of steatosis, lobular

inflammation (LI), and ballooning of livers from male Flox and F4MKO mice on 24-weeks NASH diet (Flox, $n = 8$ biologically independent animals; MKO, $n = 7$ biologically independent animals). **g**, **h** Serum ALT and AST levels in Flox and F4MKO mice on 24-weeks NASH diet ($n = 6$ biologically independent animals). All results were shown as mean ± SEM, $*p < 0.05$, $**p < 0.01$, $***p < 0.001$. A two-tailed Student t test was used for statistical analysis. LI, lobular inflammation; NAS, NAFLD activity score; ALT, alanine aminotransferase; AST, aspartate aminotransferase. Source data are provided as a Source data file.

expressed genes identified in our previous F4MKO muscle RNA-seq data[20], and identified 15 overlapping downregulated and three overlapping upregulated genes (Supplementary Fig. 3b). Next, we overlapped these 18 proteins with myokines obtained from public literature[29], identifying 12 downregulated and one upregulated myokine (Supplementary Fig. 3c, d). To further identify changes in these 13 myokines secreted by skeletal muscles, we examined these genes expression in the GAS of F4MKO NASH mouse models. We just found that Decorin, Fibronectin1, Fstl1, and Mmp2 were significantly downregulated in the GAS of F4MKO mice (Supplementary Fig. 3e). Fstl1, Mmp2, and Decorin expression was reported to be associated with NASH[23,30,31]. However, Mmp2 and Decorin exhibit a protective role on NASH, downregulation is likely to worsen NASH pathology[30,31]. Fstl1 was reported to promote fibrosis in the liver and Fibronectin 1 was expression important for the extracellular matrix[23,32]. We thus choose Fstl1 as a candidate that linked the muscle and liver tissues in our models. Meanwhile, we found that the protein expression of FSTL1 did not change in the GAS F4MKO mice fed a normal chow diet, whereas it decreased significantly in the GAS, soleus, and quadriceps of F4MKO mice under NASH model conditions (Fig. 3a, Supplementary Fig. 4a–c); the same was observed with plasma FSTL1 (Fig. 3c). Consistent with this, FSTL1

expression was higher in the GAS of NASH mice but lower in the GAS of NASH mice subjected to exercise (Fig. 3b).

We then assessed whether Fstl1 expression was transcriptionally regulated by IRF4. We constructed luciferase reporter genes containing different segments of the FSTL1 promoter and co-transfected them into 293T cells with pCDH-GFP or pCDH-IRF4. IRF4 increased the activity of the FSTL1 promoter in the −1520 to +76 region, whereas IRF4 had no effect on the activity of the promoter in the −1122 to +76 region, indicating that the element responsible for the action of IRF4 on FSTL1 promoter was positioned between bases −1520 and −1122 (Fig. 3d). Next, we analyzed the FSTL1 promoter between −1520 and −1122 using an online tool for predicting the interaction between promoters and transcription factors (http://jaspar.genereg.net). We subsequently identified three potential binding sites of IRF4 (ISRE) within this region of the FSTL1 promoter. We then constructed three mutant reporter genes with a disrupted ISRE motif, and re-conducted the dual-luciferase reporter assay. IRF4 exerted no effect on the FSTL1 promoter with the three different mutations (Fig. 3e). The data suggested that IRF4 manipulates FSTL1 expression by enhancing the activity of the FSTL1 promoter. Furthermore, the −1520 to −1122 region is responsible for this effect. Chip-qPCR results further confirmed that IRF4 was bound to the −1520 to 1122 region of FSTL1 promoter (Fig. 3f).

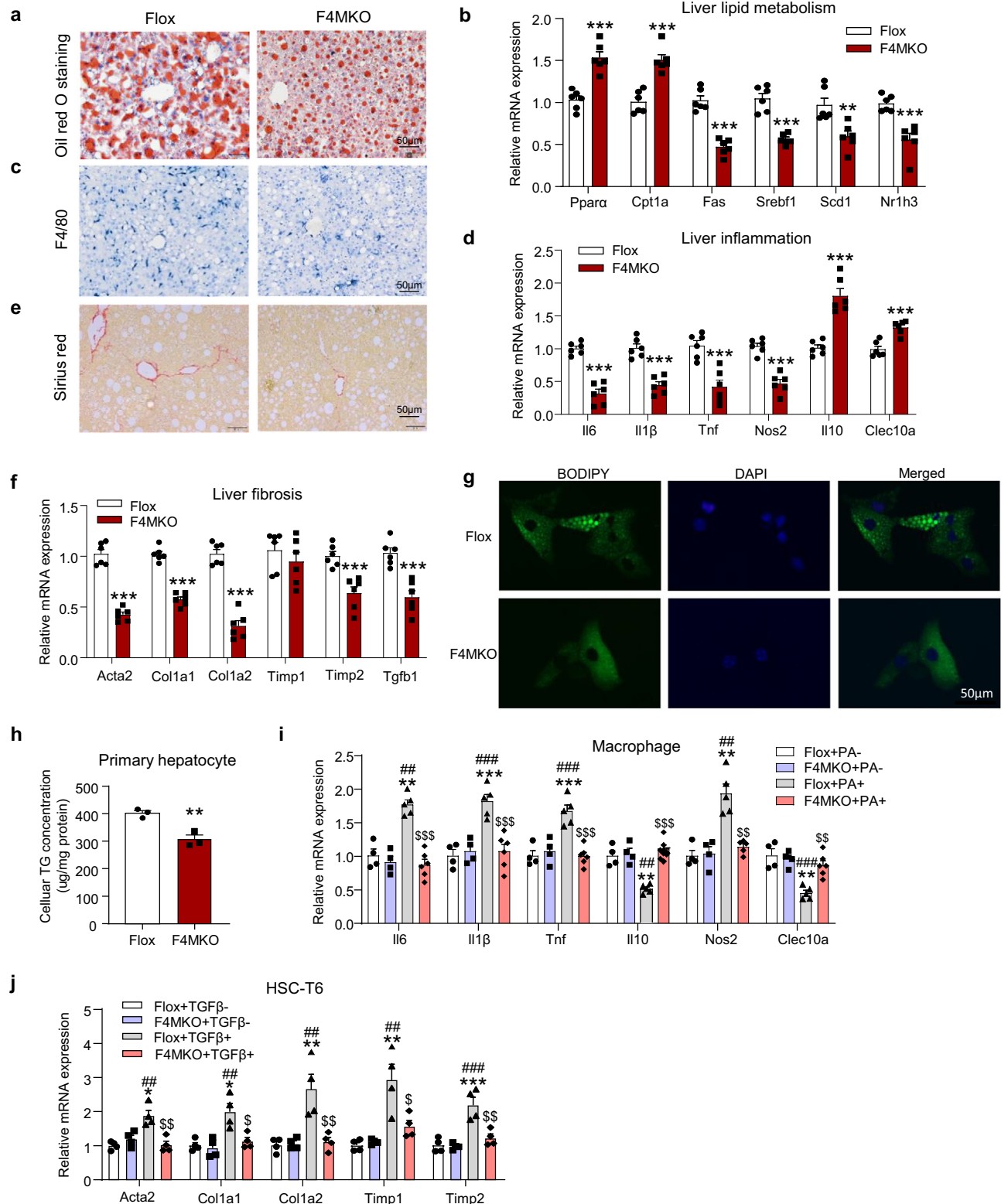

## Skeletal muscle-derived FSTL1 was responsible for IRF4-mediated NASH progression

To verify whether FSTL1 mediates IRF4's effects in liver cells in vivo, we induced the overexpression of FSTL1 directly in the GAS of mice using adenovirus-associated virus (AAV) delivery. Flox and F4MKO mice were first modeled with NASH. After 24-week NASH modeling, AAV-FSTL1, tagged with flag, was injected into the GAS. After 4 weeks of the AAV injection, the mRNA expression of FSTL1 in GAS was increased significantly in the AAV-FSTL1-treated group (Supplementary Fig. 4d), and the FSTL1 protein levels in GAS and serum also increased significantly (Fig. 4a, b). The circulating FSTL1 level also increased significantly (Fig. 4c). We observed that the FSTL1 level in the liver increased after AAV injection (Supplementary Fig. 4e), but the FSTL1 mRNA expression in the liver remained unchanged (Supplementary Fig. 4f), indicating that FSTL1 was secreted from the GAS and moved to the liver. There were no differences in the body, GAS, and liver weight

**Fig. 2 | Skeletal muscle-specific IRF4 deletion reduces lipid accumulation, inflammation, and fibrosis in liver. a** Oil red O staining of liver in Flox and F4MKO mice fed with 24-weeks NASH diet ($n = 3$ biologically independent animals. Scale bars, 50 μm). **b** Relative mRNA expression of lipid metabolism-related genes in liver of Flox and F4MKO mice on 24-weeks NASH diet ($n = 6$ biologically independent animals). **c** F4/80 staining in liver of Flox and F4MKO mice fed with 24-weeks NASH diet ($n = 3$ biologically independent animals. Scale bars, 50 μm). **d** Relative mRNA expression of inflammatory genes in Flox and F4MKO mice on 24-weeks NASH diet ($n = 6$ biologically independent animals). **e** Sirius red staining of liver in Flox and F4MKO mice fed with 24-weeks NASH diet ($n = 3$ biologically independent animals. Scale bars, 50 μm). **f** Relative mRNA expression of fibrosis genes in Flox and F4MKO mice fed with 24-weeks NASH diet ($n = 6$ biologically independent animals). **g** Immunofluorescence of primary hepatocytes co-cultured with gastrocnemius from F4MKO and Flox mice ($n = 3$ biologically independent cell experiments. Scale bars, 50 μm). Primary hepatocytes cells treated with 25 mM glucose and 200 μM PA were co-cultured with gastrocnemius from F4MKO and Flox mice for 48 h.

**h** Cellular TG level of primary hepatocytes from (**g**) ($n = 3$ biologically independent cell experiments). **i** Relative mRNA expression of inflammatory genes in macrophage cells co-cultured with gastrocnemius from Flox and F4MKO mice (Flox+PA-, $n = 4$ biologically independent cell experiments; MKO + PA-, $n = 4$ biologically independent cell experiments; Flox+PA +, $n = 5$; MKO + PA +, $n = 6$ biologically independent cell experiments). Macrophages were treated with 200 μM PA or vehicle control (**j**) Relative mRNA expression of fibrosis genes of HSC-T6 cells co-cultured with gastrocnemius from Flox and F4MKO mice ($n = 4$ biologically independent cell experiments). HSC-T6 cells were treated with 10 ng/ml rTGFβ or vehicle control. All results were shown as mean ± SEM, **$p < 0.01$, ***$p < 0.001$, compared with the Flox group, a two-tailed Student t test was used for statistical analysis (**b, d, f, h**); **$p < 0.01$, ***$p < 0.001$, compared with the Flox+PA- or Flox+TGFβ- group, ##$p < 0.01$, ###$p < 0.001$, compared with the F4MKO + PA- or F4MKO + TGFβ- group, $$p < 0.05$, $$$p < 0.01$, compared with the Flox+PA+ or Flox+TGFβ+, a two-way ANOVA followed by Bonferroni post-tests was used for statistical analysis (**i, j**). Source data are provided as a Source data file.

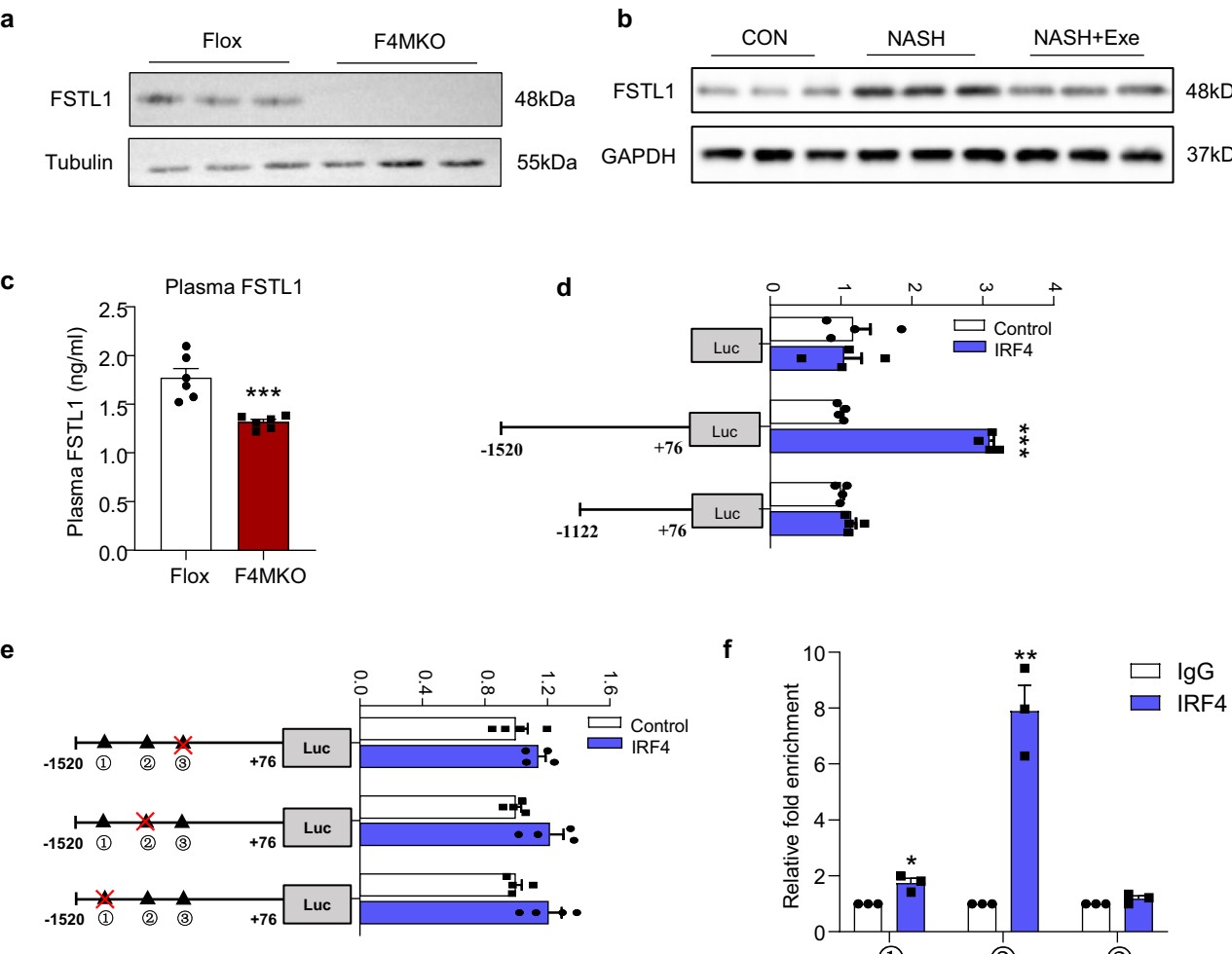

**Fig. 3 | FSTL1, a myokine, is decreased in MKO mice and transcriptionally regulated by IRF4. a** Western blot analysis of the expression of FSTL1 in skeletal muscle of Flox and F4MKO mice fed with 24-weeks NASH diet ($n = 3$). **b** Western blot analysis of the expression of FSTL1 in skeletal muscle of CON, NASH and NASH+Exe mice ($n = 3$). **c** Elisa analysis of the level of plasma FSTL1 from Flox and NASH mice fed with 24-weeks NASH diet ($n = 6$ biologically independent animals). **d** Dual Luciferase Assays of Fstl1 promoter ($n = 4$ biologically independent cell experiments). Luciferase activity was corrected for Renilla luciferase activity and normalized to Control group. **e** The mutant Fstl1 promoter fused to a luciferase reporter gene was co-transfected into 293 T cells together with pCDH-puro and

pCDH-IRF4 promoter ($n = 4$ biologically independent cell experiments). ①−1492/−1477 mutant, ②−1254/−1232 mutant, ③−1195/−1180 mutant. Luciferase activity was corrected for Renilla luciferase activity and normalized to Control group. **f** QPCR analysis of each ChIP-DNA sample was performed for FSTL1, β-ACTIN ($n = 3$ biologically independent cell experiments). Results are reported as fold enrichment of immunoprecipitated DNA from each sample relative to the DNA immunoprecipitated with the non-specific antibody, and were plotted in a scale in which the final value of IgG was arbitrarily set to 1. All results were shown as means ± SEM, *$p < 0.05$, **$p < 0.01$, ***$p < 0.001$. A two-tailed Student t test was used for statistical analysis. Source data are provided as a Source data file.

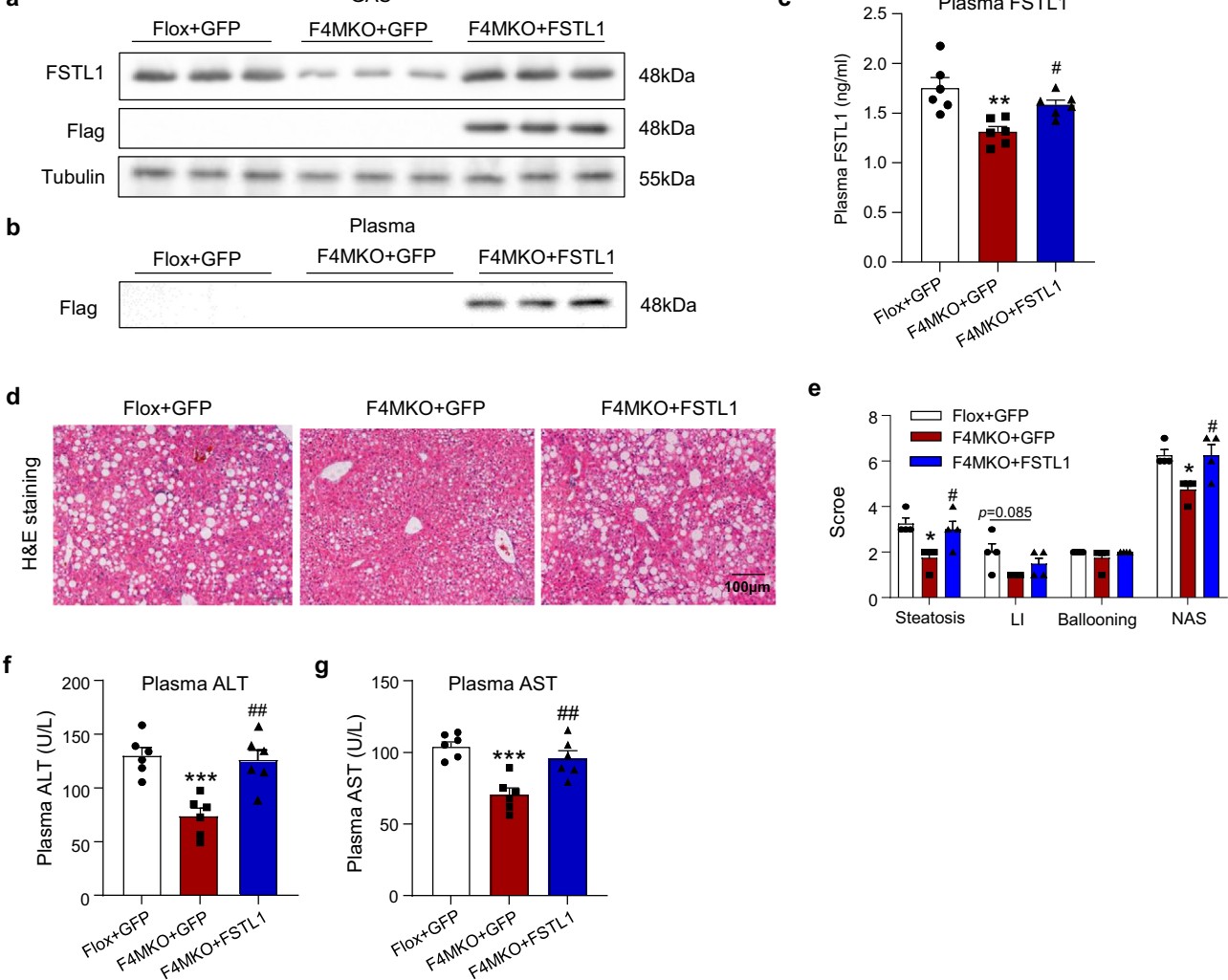

**Fig. 4 | Skeletal muscle-derived FSTL1 is responsible for IRF4-mediated NASH progression. a**, **b** Western blot analysis of the expression of FSTL1- and FLAG in skeletal muscle and FLAG in plasma of Flox+GFP, F4MKO + GFP, and F4MKO + FSTL1 mice (*n* = 3 biologically independent animals). **c** Elisa analysis of the level of plasma FSTL1 from mice of (**b**) (*n* = 6). **d** H&E staining of liver in Flox+GFP, F4MKO + GFP, and F4MKO + FSTL1 mice on 24-weeks NASH diet (*n* = 4 biologically independent animals. Scale bars, 100 µM). **e** Quantification of steatosis, lobular

inflammation (LI), and ballooning of livers from Flox+GFP, F4MKO + GFP, and F4MKO + FSTL1 mice (*n* = 4 biologically independent animals). **f**, **g** Plasma ALT, AST levels in Flox+GFP, F4MKO + GFP and F4MKO + FSTL1 mice on 24-weeks NASH diet (*n* = 6 biologically independent animals). All results were shown as means ± SEM, *$p < 0.05$, **$p < 0.01$, ***$p < 0.001$, compared with the Flox+GFP group; #$p < 0.05$, ##$p < 0.01$, compared with the F4MKO + GFP group, a two-tailed Student t test was used for statistical analysis. Source data are provided as a Source data file.

among Flox, F4MKO, and F4MKO + FSTL1 mice (Supplementary Fig. 3g–k) fed the NASH diet. Liver tissue H&E staining showed that the degree of steatosis and inflammation in F4MKO mice injected with AAV-FSTL1 were similar to that observed in Flox mice (Fig. 4d), which was further illustrated by the NAS scores of the liver tissue of Flox, F4MKO, and F4MKO + FSTL1 mice (Fig. 4e). In addition, the F4MKO + FSTL1 mice exhibited higher serum ALT and AST levels (Fig. 4f, g). These data suggested that skeletal muscle-derived FSTL1 contributed to the amelioration of hepatic pathology in F4MKO mice.

Furthermore, we induced FSTL1 knockdown in both the liver and GAS using AAV. FSTL1 knockdown in the GAS (GFSTL1-KD) did not result in any significant improvement in circulating triglyceride level, hepatic lipid accumulation, or hepatic damage (Supplementary Fig. 5a, c–e). However, we observed a mild reduction in liver fibrosis in GFSTL1-KD mice assessed by Sirius Red staining (Supplementary Fig. 5b). In addition, the expression of key genes associated with fibrosis (Acat2 and Col1a1) was also found to be lowered (Supplementary Fig. 5f). However, we did not observe any significant effects on the NASH phenotype (Supplementary Fig. 5g–k) upon FSTL1 knockdown in the liver. These results provided further confirmation that

FSTL1 derived from skeletal muscle plays a dominant role in the NASH pathology phenotype.

### FSTL1 overexpression of counteracted the effects on the liver induced upon the ablation of skeletal muscle IRF4

To test whether FSTL1 mediates the effect of disrupted IRF4 expression in skeletal muscles on NASH progression, we performed the transcriptome analysis of liver tissues in Flox, F4MKO, and F4MKO + FSTL1 mice. KEGG pathway analysis revealed that the non-alcoholic fatty liver disease (NAFLD) pathway was repressed in F4MKO mice compared with that in control mice. Upon injection with AAV-FSTL1, the NAFLD pathway was activated relative to that in AAV-GFP-injected F4MKO mice (Fig. 5a, b). Next, we verified that skeletal muscle-derived FSTL1-mediated NASH progression with regards to hepatic lipid metabolism, fibrosis, and inflammation in vivo when specific IRF4 deletion was induced in skeletal muscles. First, we found that triglyceride content was significantly lowered in the liver of F4MKO mice but increased in the liver of AAV-FSTL1-injected F4MKO mice (Fig. 5c, Supplementary Fig. 6a). Consistent with this, key genes associated with lipogenesis were downregulated in F4MKO mice, but their expression

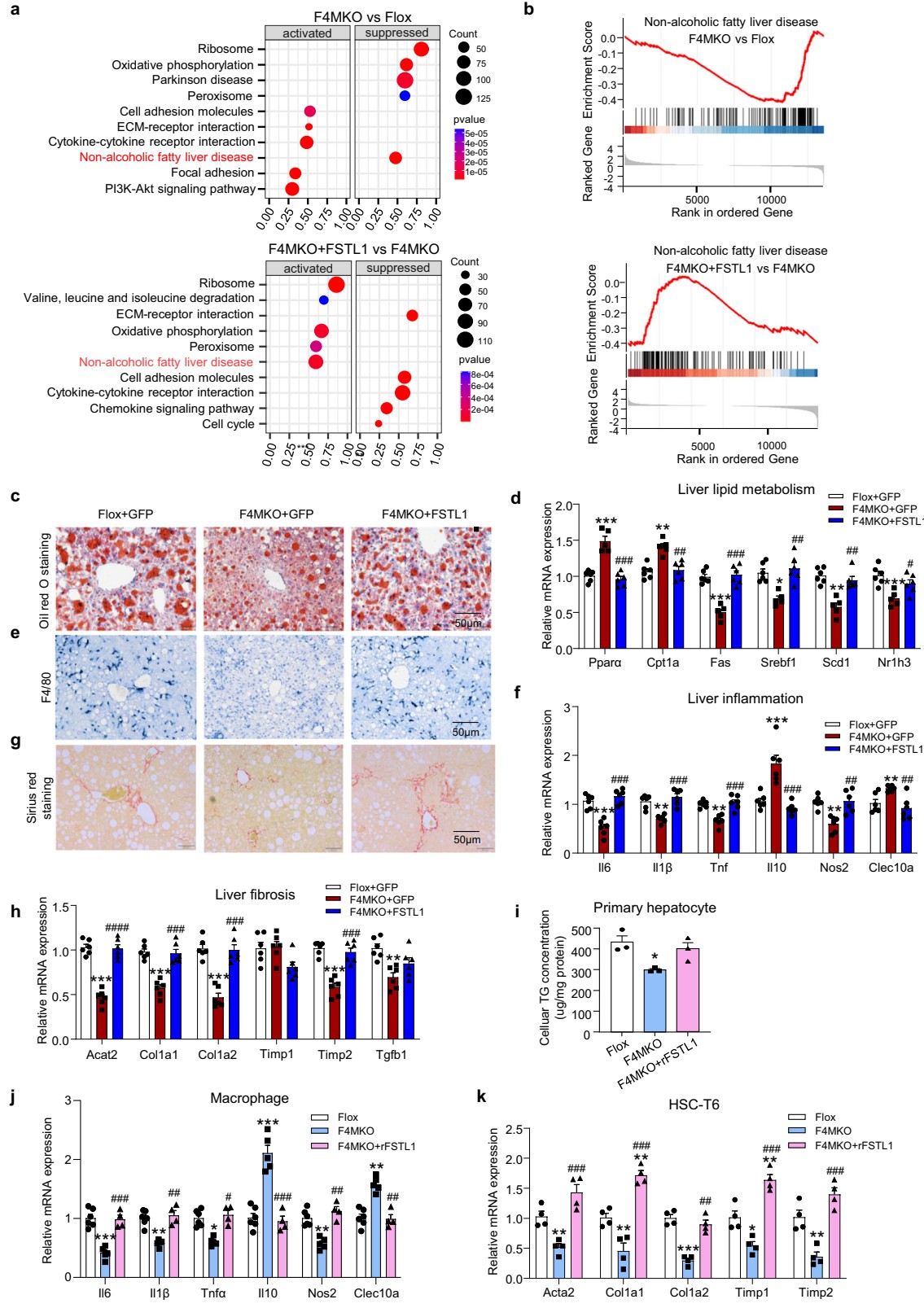

was restored in F4MKO + FSTL1 mice (Fig. 5d). Second, we checked the degree of inflammation in the liver. F4/80 staining showed that there were less kupffer cells in F4MKO than in control mice, whereas the number of cells was greater in F4MKO mice injected with AAV-FSTL1 (Fig. 5e). Moreover, the expression of inflammatory genes, such as Il-6, Il-1β, Tnf, and Il-10 was reversed in F4MKO mice injected with AAV-FSTL1 (Fig. 5f). Third, Sirius Red staining showed that the degree of

fibrosis in the liver was similar to that in control and F4MKO + FSTL1 groups but decreased in the F4MKO group (Fig. 5g). QPCR analysis revealed the same changes in the expression patterns of fibrosis-related genes (Fig. 5h).

In addition, recombinant protein-FSTL1 (rFSTL1) increased the TG concentration in primary hepatocytes and HepG2 cells treated with PA, but not the basal levels (Supplementary Fig. 6b, c). Since FSTL1 was

**Fig. 5 | Overexpression of FSTL1 counteracts the liver effects induced by ablation of skeletal muscle IRF4. a, b** Gene set enrichment analysis of Flox, F4MKO, and F4MKO + FSTL1 liver from RNA-seq data. **c** Oil red O staining of liver in Flox+GFP, F4MKO + GFP, and F4MKO + FSTL1 mice (*n* = 3 biologically independent animals. Scale bars, 50 μm). **d** Relative mRNA expression of lipid metabolism-related genes in NASH liver of Flox+GFP, F4MKO + GFP, and F4MKO + FSTL1 mice (Flox+GFP, *n* = 6 biologically independent animals; F4MKO + GFP, *n* = 5 biologically independent animals; F4MKO + FSTL1, *n* = 6 biologically independent animals). **e** F4/80 staining in NASH liver of Flox+GFP, F4MKO + GFP, and F4MKO + FSTL1 mice (*n* = 3 biologically independent animals. Scale bars, 50 μm). **f** Relative mRNA expression of inflammatory genes in NASH liver of Flox+GFP, F4MKO + GFP, and F4MKO + FSTL1 mice (*n* = 6 biologically independent animals). **g** Sirius red staining of NASH liver in Flox+GFP, F4MKO + GFP, and F4MKO + FSTL1 mice (*n* = 3 biologically independent animals. Scale bars, 50 μm). **h** Relative mRNA expression of fibrosis genes in NASH liver of Flox+GFP, F4MKO + GFP, and F4MKO + FSTL1 mice (*n* = 6 biologically independent animals). **i** Cellular TG level of primary hepatocytes co-cultured with indicated gastrocnemius (*n* = 3 biologically independent cell experiments). Primary hepatocytes cells treated with 25 mM glucose and 200 μM PA were co-cultured with gastrocnemius from Flox or F4MKO mice for 48 h. **j** Relative mRNA expression of inflammatory genes in macrophages co-cultured with gastrocnemius (Flox+GFP, *n* = 6 biologically independent cell experiments; F4MKO + GFP, *n* = 5 biologically independent cell experiments; F4MKO + FSTL1, *n* = 4 biologically independent cell experiments). Macrophages were treated with 200 μM PA for 24 h, then macrophages were co-cultured with GAS from Flox and F4MKO mice for 48 h and treated with or without 100 ng/ml rFSTL1. **k** Relative mRNA expression of fibrosis genes in HSC-T6 cells co-cultured with GAS (*n* = 4 biologically independent cell experiments). HSC-T6 cells were treated with 10 ng/ml rTFGβ for 24 h, then HSC-T6 cells were co-cultured with GAS from Flox and F4MKO mice for 48 h and treated with or without 100 ng/ml rFSTL1. All results were shown as mean ± SEM. \**p* < 0.05, \*\**p* < 0.01, \*\*\**p* < 0.001, compared with the Flox+GFP group; #*p* < 0.05, ##*p* < 0.01, ###*p* < 0.001, compared with the F4MKO + GFP group, a two-way ANOVA followed by Bonferroni post-tests was used for statistical analysis. Source data are provided as a Source data file.

---

downregulated in F4MKO mice, we added rFSTL1 into the co-culture system. In the primary hepatocyte steatosis cell model, rFSTL1 restored the cellular TG levels and lipid accumulation in cells co-cultured with GAS from F4MKO mice (Fig. 5i). rFSTL1 exerted the same effect in an HepG2 steatosis cell model (Supplementary Fig. 6d). Meanwhile, we tested the effects of rFSTL1 in macrophage cells (Supplementary Fig. 6e–h). rFSTL1 counteracted inflammatory gene expression in macrophages co-cultured with GAS from F4MKO mice (Fig. 5j). Consistent with this, rFSTL1 reversed liver fibrosis gene expression in HSC-T6 cells co-cultured with GAS from F4MKO mice (Fig. 5k, Supplementary Fig. 6i–m). In conclusion, FSTL1 overexpression could counteract the effects of skeletal muscle IRF4 ablation in steatosis, fibrosis, and inflammation in NASH mice.

### Skeletal muscle IRF4 regulated liver metabolism via FSTL1-DIP2A/CD14

To screen the receptors of FSTL1 that is/are responsible for the liver phenotypes in F4MKO mice, we measured the expression of their transcripts. Twelve receptors were reported to be related to FSTL1[33–36]. According to sc-RNA-Seq data from human and mouse livers, these 12 receptors were expressed in different patterns in different cell types (Supplementary Fig. 7a–d)[37–40]. We then measured the expression of these receptors in our models. The expression of seven receptors was altered in F4MKO mice; these were including Cd14, Dip2a, Eng, Bmpr2, Tgfbr1, Tgfbr2, and Bmpr1b (Supplementary Fig. 7e). Given that the Bmpr1b transcript was not detected in mouse NASH liver sc-RNA-Seq (Supplementary Fig. 7d), we designed the shRNA of the remaining six receptors to further clarify how FSTL1 regulates liver metabolism. We used mouse AML12 hepatocytes, hepatic stellate HSC-T6 cell, and a macrophage cell line, which were infected with an shRNA lentivirus to knockdown the receptors of FSTL1 (Supplementary Figs. 8a, 9a, 10a). Surprisingly, rFSTL1 increased lipid accumulation in AML12 cells, but the effects were only blocked by shDip2a and shCd14 and not by other receptors (Supplementary Fig. 8b–g). The blockade of Dip2a and Cd14 in AML12 cells did not result in any changes in the expression of the other receptors (Supplementary Fig. 8h, i). Again, we found that inflammatory gene expression was increased in macrophage cells treated with rFSTL1, whereas the expression of these genes normalized when the cells were infected with shCd14 (Supplementary Fig. 9b–g). Similarly, when Cd14 knockdown in macrophage cells did not cause changes in other receptors (Supplementary Fig. 9h). Combining these findings with the sc-RNA-Seq data, we selected four to six receptors (Eng, Bmpr2, Tgfbr2, and Dip2a) that showed a change in expression patterns in NASH HSCs. The fibrosis-related genes were upregulated in rFSTL1 supplement cells but downregulated only in shDip2a infected cells (Supplementary Fig. 10b–e). However, the expression of other receptors did not change in HSC-T6 cells infected with shDip2a (Supplementary Fig. 10f).

To identify the receptor that contributes to the effects of the skeletal muscle IRF4-FSTL1 axis on the liver, co-culture experiments were carried out. FSTL1 could not restore cellular lipid accumulation and lipid metabolism gene expression in Dip2a or Cd14 knockdown AML12 steatosis cells co-cultured with GAS from F4MKO mice (Fig. 6a–h). Moreover, following Cd14 knockdown in macrophages and Dip2a knockdown in HSC-T6, rFSTL1 was unable to reverse inflammation-related gene expression in macrophages and fibrosis-related gene expression in HSC-T6 cells co-cultured with GAS from F4MKO mice (Fig. 6i–r). Collectively, these data suggest that skeletal muscle IRF4 regulates lipid metabolism, fibrosis, and inflammation via FSTL1-DIP2A/CD14. To further validate these results in vivo, we used AAV delivery to directly block both Dip2a and Cd14 expression in the liver inducing the overexpression of FSTL1 in the GAS. FSTL1 could not rescue hepatic damage, lipid accumulation, and fibrosis when both Dip2a and Cd14 were knocked down in the liver (Fig. 6s–w). In summary, these data revealed that skeletal muscle IRF4 regulates NASH progression through the FSTL1-DIP2A/CD14 pathway.

### The serum FSTL1 level was associated with hepatic steatosis and fibrosis in humans

To determine whether the serum FSTL1 level associated with liver pathology in humans, we analyzed specimens from 43 healthy participants, 43 patients with steatosis, 41 patients with early NASH and 53 with advanced NASH. The demographic characteristics are presented in Fig. 7a. Although the FSTL1 level did not vary by age, BMI, plasma TG level, free fatty acid (FFA) levels, or cholesterol levels, we observed a significant positive correlation between FSTL1 expression and liver pathologies (steatosis, ballooning, fibrosis, and NAS) (Fig. 7b–d). Notably, the plasma FSTL1 level was also correlated with glucose levels or HbA1c (Fig. 7e–g). Nonetheless, the Fstl1 mRNA level in liver biopsies from patients with NASH was significantly lower than that in controls (Fig. 7h, i). This, suggested that the increased plasma FSTL1 level in patients with NASH might be owing to FSTL1 secretion from other tissues, such as skeletal muscles, as shown in the present study. Consistent with the level of FSTL1 in the liver, the expression of its receptors was decreased in the liver from patients with NASH in our study as well as in other studies[41] (Fig. 7h, i, Supplementary Fig. 11). Our data thus strongly suggested a model in which skeletal muscle IRF4 transcriptionally regulates FSTL1, and the latter is secreted in the liver to mediate liver metabolism through DIP2A/CD14, eventually affecting the pathology of NASH (Fig. 7j).

## Discussion

The association between the pathogenesis and natural course of NASH and skeletal muscle dysfunction has been recognized widely in recent times. Here, we showed that skeletal muscle plays a role in the progression of NASH via the IRF4-FSTL1-DIP2A/CD14 axis (Fig. 7j).

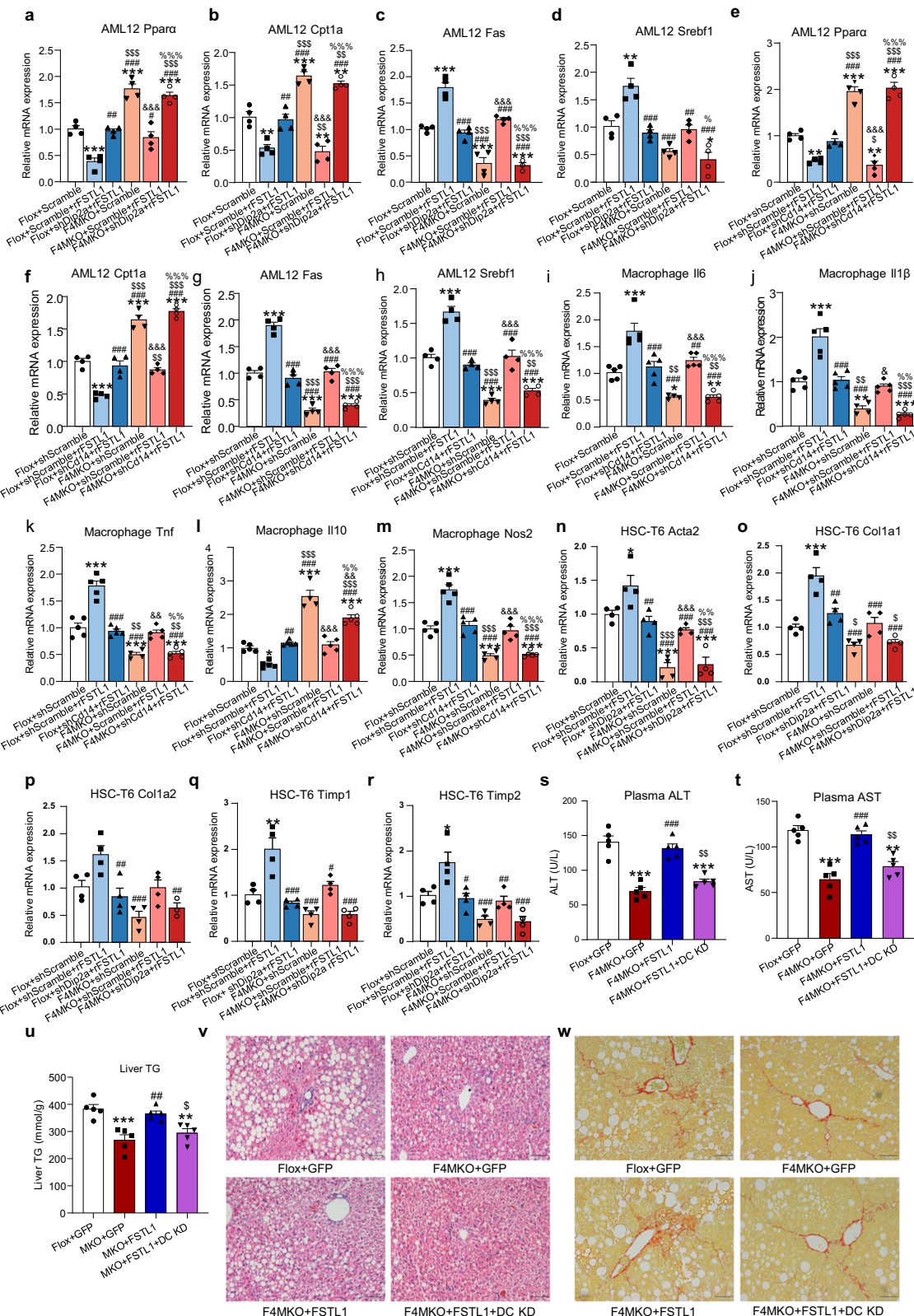

Surprisingly, IRF4 expression was higher in skeletal muscles, but not in the liver tissues, in NASH models. IRF4-specific deletion in skeletal muscle alleviated steatosis, inflammation, and fibrosis in the liver. These effects were mediated by a myokine named FSTL1, which exerts its function through DIP2A and CD14 in the liver. Importantly, serum FSTL1 expression is positively correlated with NASH progression in humans. Skeletal muscles are a major reservoir for secreted FSTL1, as

addressed in other studies[27]. Our findings suggest a novel pathway that plays a role in the communication between skeletal muscles and the liver, which are potential drug targets in metabolic diseases.

Of note, IRF4 was shown to play a key role in the development and function of immune cells[14,42]. We have described that IRF4 expression in fat induces catabolic effects (increased lipolysis, anti-lipogenesis, and increased M2 macrophage abundance)[15,43], and the overexpression

**Fig. 6 | Skeletal muscle IRF4 regulates liver metabolism via FSTL1-DIP2A/CD14.**
**a–h** Lipid metabolism genes of AML12 cells co-cultured with GAS from Flox or
F4MKO mice (**c** F4MKO+shDip2a + rFSTL1, *n* = 3 biologically independent cell
experiments, *n* = 4 biologically independent cell experiments for others). AML12
cells co-cultured with GAS from Flox or F4MKO mice, were infected with shDip2a or
shCd14 lentivirus and then treated with 200 μM PA for 24 h. In shScramble+rFSTL1
and shDip2a + rFSTL1 or shCd14 + rFSTL1 group, AML12 cells were additionally
treated with 100 ng/ml rFSTL1. **i–m** Relative mRNA expression of inflammatory
genes in macrophages co-cultured with GAS from Flox and F4MKO mice
(**i–m** F4MKO+shScramble, *n* = 4 biologically independent cell experiments, *n* = 5
biologically independent cell experiments for others). Macrophages, co-cultured
with GAS from Flox and F4MKO mice, were infected with shDip2a lentivirus and
then treated with 200 μM PA for 24 h. In shscramble+rFSTL1 and shDip2a + rFSTL1
group, macrophages were additionally treated with 100 ng/ml rFSTL1. **n–r** Relative
mRNA expression of fibrosis genes in HSC-T6 cells co-cultured with GAS from Flox
and F4MKO mice (**p** F4MKO+shDip2a + rFSTL1, *n* = 3 biologically independent cell
experiments, *n* = 4 biologically independent cell experiments for others). HSC-T6
cells co-cultured with GAS from Flox and F4MKO mice, were infected with shDip2a
lentivirus and then treated with 10 ng/ml TGFβ1 for 24 h. In shscramble+rFSTL1 and

shDip2a + rFSTL1 group, HSC-T6 cells were additionally treated with 100 ng/ml
rFSTL1. **s, t** Plasma ALT and AST levels in Flox+GFP, F4MKO + GFP, F4MKO + FSTL1,
and F4MKO + FSTL1 + DC KD (Dip2a and Cd14 knock down) mice on 24-weeks
NASH diet (*n* = 5 biologically independent animals). **u** Liver TG level in Flox+GFP,
F4MKO + GFP, F4MKO + FSTL1, and F4MKO + FSTL1 + DC KD mice (*n* = 5 biologi-
cally independent animals). **v, w** H&E staining and Sirius red staining of NASH liver
in Flox+GFP, F4MKO + GFP, F4MKO + FSTL1, and F4MKO + FSTL1 + DC KD mice
(*n* = 3 biologically independent animals. Scale bars, 50 μm). All results are shown as
means ± SEM. *$p < 0.05$, **$p < 0.01$, ***$p < 0.001$, compared with the Flox+shScram-
ble group; #$p < 0.05$, ##$p < 0.01$, ###$p < 0.001$, compared with the Flox+shScramble+
rFSTL1 group; $$p < 0.05$, $$$p < 0.01$, $$$$p < 0.001$, compared with the Flox+
shDip2a + rFSTL1 group or Flox+shCd14 + rFSTL1 group; &$p < 0.05$, &&$p < 0.01$,
&&&$p < 0.001$, compared with the F4MKO+shScramble group; %$p < 0.05$, %%$p < 0.01$, %
%%$p < 0.001$, compared with the F4MKO+shScramble+rFSTL group. A two-way
ANOVA followed by Bonferroni post-tests was used for statistical analysis (**a–r**).
**$p < 0.01$, ***$p < 0.001$, compared with the Flox+GFP group; ###$p < 0.001$, compared
with the F4MKO group; $$p < 0.05$, $$$p < 0.01$, compared with the F4MKO + FSTL1
group. The one-way ANOVA followed by Bonferroni post-tests was used for sta-
tistical analysis (**s, t**) Source data are provided as a Source data file.

---

of IRF4 in skeletal muscle favors diet induced-obesity[20]. Recently, we
addressed that F4MKO mice are obesity-resistant and exhibit
enhanced energy expenditure, insulin sensitivity, and exercise
capacity[19,20]. Weight loss can help alleviate lipid-induced hepatocellular
injury by mobilizing fat from the liver. However, limited information is
available about such phenomena in the absence of weight loss, espe-
cially with respect to alterations in hepatic fat metabolism. Here,
F4MKO mice fed the NASH diet showed an improvement in hepatic
injury without body weight changes. Upon further investigation, IRF4
was found to transcriptionally regulate FSTL1 expression in skeletal
muscle, the latter being secreted in the liver to exert its function on
liver cells.

Skeletal muscles act as endocrine organs, secreting various
myokines that affect other metabolic tissues, including the liver[7]. The
insufficient or excessive secretion of myokines may regulate steatosis,
inflammation, or fibrosis in the liver. FSTL1 is a myokine that mediates
the function of IRF4s function in the liver. The serum FSTL1 level is
positively correlated with NASH in humans. rFSTL1 treatment increases
lipid accumulation in hepatocytes, enhances fibrosis in HSCs, and
augments inflammatory gene expression in macrophages. However,
FSTL1 expression is reported to be induced by dynamic resistance
exercise and promotes cardiac rehabilitation in MI rats[26]. The con-
troversial function of FSTL1 may be attributed to the post-translational
modification of FSTL1. For example, in the sequence of mouse Fstl1,
three potential sites are present for N-glycosylation and two potential
sites are present for O-glycosylation. Glycosylation at these sites leads
to cell-type specificity[44]. The activation of various signaling pathways
through different receptors is likely responsible for that: the protective
role of FSTL1 in the heart is mediated *via* DIP2A in MI rats[26]. In this
study, not only DIP2A but also CD14 was found to responsible for the
role of FSTL1 in the liver. However, the fact that additional endogenous
or exogenous factors are involved in the regulation process cannot be
ignored.

In addition, the circulating levels of FSTL1 were found to be ele-
vated in patients with various diseases, such as cardiovascular
diseases[35,45,46], and liver disease[22,41]. In line with previous findings, our
findings showed that FSTL1 concentrations were increased in patients
with NASH. However, the mRNA levels of Fstl1 in the liver of patients
with NASH were found to be lowered in our study, which was con-
sistent with previous findings from transcriptome sequencing
experiments[41]. Collectively, our findings, combined with previous
results, suggest that muscle-derived FSTL1 is involved in the regulation
of NASH development. This hypothesis is further supported by the
following lines of evidence from in vitro and in vivo experiments. First,
FSTL1 knockdown in the liver did not affect NASH progression.

Second, FSTL1 knockdown in muscle partially alleviated NASH phe-
notypes, indicating that muscle-derived FSTL1 plays an essential role in
the development of NASH. Third, findings from in vitro studies showed
that recombinant FSTL1 protein could rescue the steatosis cell phe-
notype. Finally, in our model, replenishing FSTL1 in muscle effectively
restored the NASH phenotypes induced by the reduction of FSTL1
expression resulting from IRF4 knockout in muscle. Thus, muscle-
derived FSTL1 plays a major role in the regulation of NASH. In con-
clusion, our findings demonstrate that IRF4 expressed in skeletal
muscle plays a critical role in the pathogenesis of NASH through the
FSTL1-DIP2A/CD14 pathway. Targeting this signaling pathway could
serve as an attractive strategy for NASH management.

However, there are several limitations of this study. First, our
study used high cholesterol content (1.25%) in the NASH diet, whether
other NASH diets can obtain the same conclusion is worthy of further
investigation. Second, we knockdown FSTL1 only in GAS but no other
skeletal muscles, which may be not enough to mimic the effect of Mly1-
cre/FSTL1. Third, we investigated the effect of FSTL1 on aggravated
NASH induced by IRF4 muscle knockout mice, but the precise mole-
cular mechanism of how FSTL1-DIP2A/CD14 pathway regulates liver
metabolism need to be investigated in the future.

## Methods
### Animals
Mice were maintained under a 12-h light/12-h dark cycle at constant
temperature (23 °C) and humidity (50–60%) with free access to food
and water. The muscle-specific IRF4 knockout mice were generated as
previous reported, detailed protocols and information regarding the
establishment have been described previously[19]. All animals received
humane care according to the criteria outlined in the Guide for the
Care and Use of Laboratory Animals. The animal experiments abide by
the ARRIVE guidelines. All animal studies were approved by the Insti-
tutional Animal Care and Use Committee of Shanghai University of
Sport (102772022DW019).

### Animal experiment design
C57BL/6 mice were obtained from Gempharmatech Co., Ltd (Nanjing,
China). The mice were allowed to acclimate to the living environment
for 7 days prior to the start of all experiments.

**Animal experiment design 1#.** The 8-week-old Wild-type male mice
(total 16), five of them were used as normal controls (con), the others
were established a NASH model by feeding the mice with a high fat and
high carbohydrate diet (42% kcal/fat, 42.7% kcal/carbohydrate, 15.2%/
protein, 41% wt sucrose, 7.5% wt maltodextrin, 3.5% wt mineral mix)

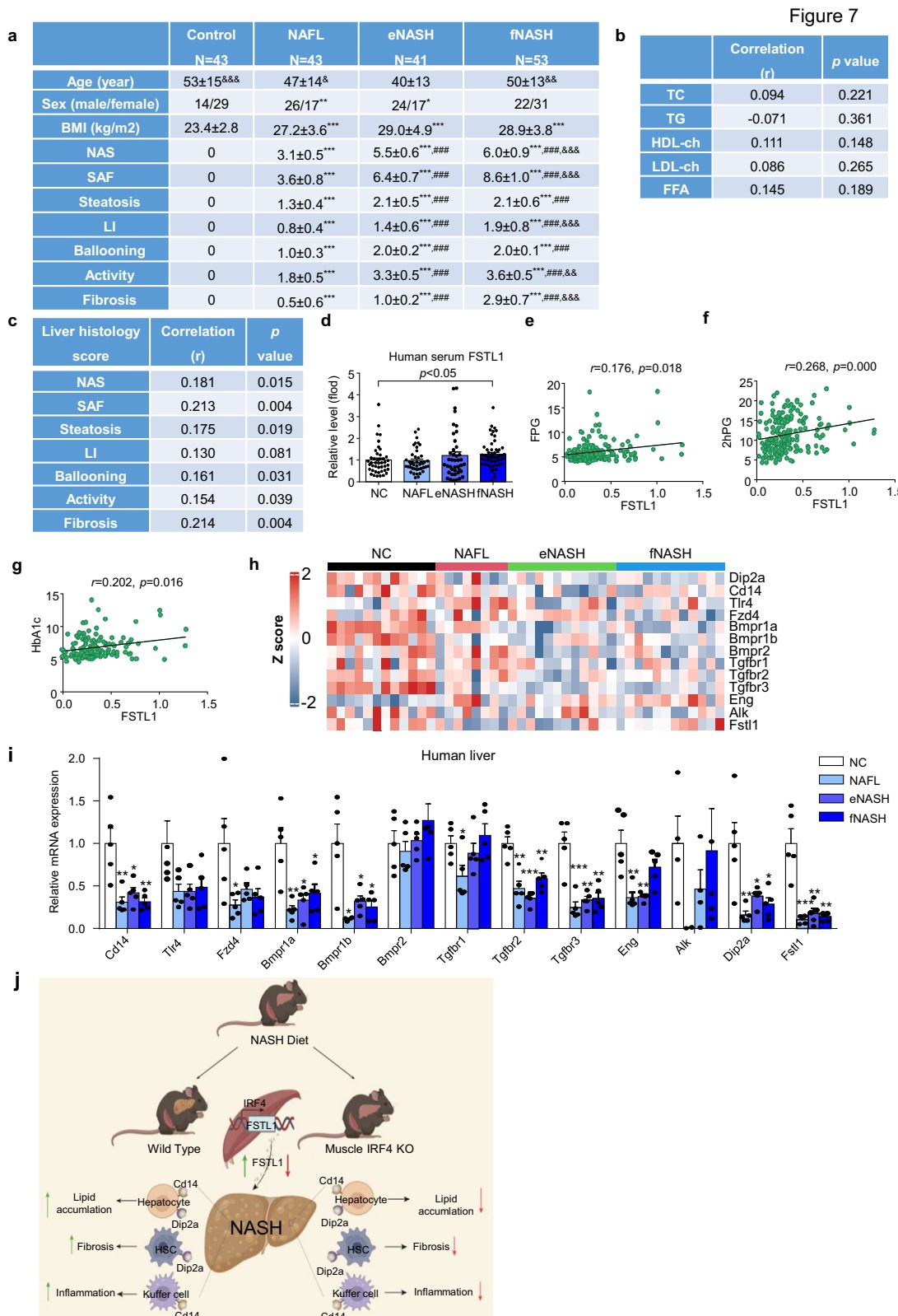

Figure 7

and containing 1.25% wt cholesterol (TD 120528, Teklad Custom Diet) with a high fructose-glucose solution (23.1 g/L d-fructose +18.9 g/L d-glucose), as previously described[47–49]. After 24-week of treatment, NASH mice were randomly divided into 2 groups, NASH group (NASH) and NASH combined with exercise group (NASH + Exe). NASH + Exe group mice were underwent 8 weeks of moderate intensity exercise intervention as previous reported[50]. After 8-weeks exercise treatment,

livers and gastrocnemius were collected from these mice to perform histological and molecular biological analysis.

**Animal experiment design 2#.** To investigate whether ablation of IRF4 in skeletal muscle would ameliorate NASH progression, we generated skeletal muscle-specific IRF4 knockout (F4MKO) mice. The male and female Flox and F4MKO mice were established NASH model for

**Fig. 7 | Serum FSTL1 level is associated with NASH progression in humans.**
**a** Clinical characteristics of subjects. **b** Associations between serum FSTL1 and serum lipid profile. **c** Associations between serum FSTL1 and liver histology score. **d** FSTL1 relative level in human serum (NC, $n = 44$ biologically independent samples; NAFL, $n = 44$ biologically independent samples; eNASH, $n = 42$ biologically independent samples; fNASH, $n = 53$ biologically independent samples). **e**–**g** Associations between serum FSTL1 and FPG, 2HPG and HbA1c. **h** Heatmap of Fstl1 and its receptors from human liver biopsy RNAseq. **i** Relative mRNA expression of FSTL1 receptors in human NASH liver ($n = 5$ biologically independent samples). **j** Working model. Figure created using BioRender.com. All results were expressed as means ± SEM. $*p < 0.05$, $**p < 0.01$, $***p < 0.001$, compared with the NC group; $###p < 0.001$, compared with the NAFL group; $\&\&p < 0.01$, $\&\&\&p < 0.001$, compared with the eNASH group; the One-way ANOVA followed by Bonferroni post-tests was used for statistical analysis (**a**). $*p < 0.05$, $**p < 0.01$, $***p < 0.001$, compared with the NC group, the One-way ANOVA was used for statistical analysis. Source data are provided as a Source data file.

24-week. After the NASH modeling period, the liver tissues were collected for the examination of diet-induced NASH in Flox and F4MKO mice.

**Animal experiment design 3#.** To specifically overexpressed FSTL1 in GAS in vivo experiments, the adeno-associated virus serotype 8 (AAV8)-muscle creatine kinase (MCK) encoding full-length FSTL1 sequences (AAV-FSTL1) delivery system was established. After 24-weeks NASH modeling, AAV-MCK promoter-FSTL1 tagged with Flag, was injected to gastrocnemius. After 4-weeks AAV injection, the liver tissue samples were collected from mice to detect corresponding histological, biochemical and molecular biological analysis.

**Animal experiment design 4#.** To construct a special knockdown of FSTL1 in vivo, AAV packaged a short hairpin RNA target FSTL1 was generated with a standard molecular procedure. And injected into male NASH mice via tail vein or directly into the gastrocnemius to achieve FSTL1 knockdown in liver or gastrocnemius, respectively. After 4-weeks AAV injection, the blood and liver tissues were collected to assess the effect of FSTL1 knockdown in the liver and muscle on the progression of NASH.

**Animal experiment design 5#.** To further validate that skeletal muscle IRF4 regulates lipid metabolism, fibrosis, and inflammation via FSTL1-DIP2A/CD14 in vivo, we constructed a special knockdown of Dip2a and Cd14 in vivo as described in model 4#. And the AAV of Dip2a and Cd14 were injected into male F4MKO mice via tail vein, based on over-expressed of FSTL1 in the GAS. After recovery for 4 weeks, the blood and liver tissues were subjected to histological analysis and biochemical analysis.

**Human subjects**
All subjects were from Zhongshan Hospital, Fudan University. A liver biopsy was performed according to the EASL−EASD−EASO clinical practice guidelines to evaluate the severity of liver histology[51]. NAFLD was histologically diagnosed by the presence of ≥5% hepatic steatosis; NAFL was histologically diagnosed by the presence of steatosis without ballooning or lobular inflammation; NASH was histologically diagnosed by the joint presence of steatosis, ballooning, and lobular inflammation[51]. The NAFLD activity score (NAS) and the steatosis, activity, and fibrosis score (SAF) were used to quantify the severity of liver steatosis, ballooning, inflammation, and fibrosis based on liver histology[52,53]. NASH with fibrosis score <2 was defined as early NASH (eNASH) and NASH with fibrosis score ≥2 was defined as fibrotic NASH (fNASH)[51]. The competing etiologies (chronic viral hepatitis, hypo-thyroidism, excessive alcohol consumption, drugs leading to steatosis) of steatosis were ruled out based on laboratory examination and liver histology. Subjects with (1) age <17 years old; (2) type 1 diabetes mellitus (T1DM), gestational diabetes, and other specific types of diabetes; (3) acute complications of diabetes; (4) severe renal disease or abnormal renal function (Cr≥115 μmol/L); or (5) history of malignant tumor, severe mental illness, or parenteral nutrition were excluded.

Finally, a total of 180 adult subjects (43 control, 43 NAFL, 41 eNASH, and 53 fNASH subjects) were included in this study. All subjects underwent routine physical measurements, blood biochemical tests, and medical history records as previously described[54]. All protocols were performed in accordance with the Declaration of Helsinki of 1975 and approved by the ethics committee of the Zhongshan Hospital, Fudan University, and each subject provided written informed consent.

**Liver histology**
Liver tissues were fixed in a 4% paraformaldehyde solution for 24 h, embedded in paraffin. Hematoxylin-eosin (H&E) (hematoxylin, E607317-0500; eosin, E607321-0100, Sangon Biotech, Shanghai, China) staining was performed on paraffin-embedded tissues to visualize the pattern of lipid accumulation. Oil Red O (E607319-0010; Sangon Biotech, Shanghai, China) staining was performed on optimal cutting temperature (OCT) compound (4583, Sakura, Torrance, CA)-embedded frozen liver sections to visualize lipid droplet accumulation. Sirius red (PH1098; Scientific Phygene, Beijing, China) staining was performed to detect liver fibrosis with standard techniques. F4/80 (Proteintech) staining was performed to detect liver lobule inflammation. Moreover, to characterize the hepatocytes lipid accumulation, bodipy (Sigma) immunofluorescence of primary hepatocytes, HepG2, and AML12 cells was performed. Histological features were observed and captured with a light microscope (OLYMPUS DP80, Olympus, Tokyo, Japan).

**Cell lines, culture conditions, and transfection**
Human HEK 293T cells, human HepG2 cells, Rat HSC-T6 cells and mouse RAW264.7 macrophage cells were cultured in DMEM/H (Gibco) with 10% fetal bovine serum (FBS, Gibco), 2 mM L-glutamine (Gibco), and 1% penicillin-streptomycin (P/S, Gibco), in a 5% $CO_2$ atmosphere at 37 °C. Mouse AML12 were grown in DMEM/F12 with 10% FBS, 1% P/S, and 1% ITS Liquid Media Supplement (Gibco, 51300044) in a 5% $CO_2$ atmosphere at 37 °C. Polyethylenimine (PolySciences, 23966) was used for transfecting according to the instructions of manufacturer.

**Isolation of primary hepatocytes**
Primary hepatocytes were isolated by the collagenase perfusion method as described previously[55]. The isolated hepatocytes were enriched by Percoll gradient centrifugation. Then cultured in William's E Medium containing 10% FBS and antibiotics on collagen-coated plates. After overnight incubation (16 h), the culture medium was refreshed.

**Protein extraction and western blot analysis**
Proteins were extracted with RIPA buffer (Beyotime Biotechnology, China)) containing protease and phosphatase inhibitors (Beyotime Biotechnology, China)) and were quantified using Rapid BCA Protein Assay Kit (Thermo Fisher, USA) according to the manufacturer's instructions. Western blot analysis was performed as previously described[19]. Briefly, 40 μg lysate was loaded onto SDS-PAGE gels, blotted onto polyvinylidene difluoride (PVDF) membranes (Millipore, USA), and incubated with antibodies. The primary antibodies used in this work including anti-IRF4 (Proteintech, 11247-2-AP, dilution 1:1000), anti-FSTL1 (Abcam, ab71548, dilution 1:1000), anti-β-Tubulin (Abclonal, AC008, dilution 1:5000), anti-GAPDH (Abways, AB0037, dilution 1:5000), anti-flag (Abclonal, AE005, dilution 1:2000).

## Analysis of gene expression by quantitative real-time PCR

TRIzol method (Thermo Fisher) was used to extract total RNA from tissues and cells according to the manufacturer's instructions. Briefly, 1 μg RNA was converted into cDNA using High-Capacity cDNA Reverse Transcription Kit (Thermo Fisher), and QPCR was performed with a QuantStudio™ 7 Flex Real-Time PCR System (Thermo Fisher) using SYBR Green PCR Master Mix (Accurate Biology) according to the manufacturer's instructions. Tbp or18s was used as the endogenous control.

## Plasmids construction

Encoding small hairpin RNA targeting FSTL1 receptor, was constructed by inserting small hairpin RNA sequence into the AgeI/EcoRI site of pLKO.1-puro (Addgene), detailed shRNA sequences are shown in Supplementary Table 1.

## AAV-FSTL1 production, purification, and injection

AAV8-MCK promoter-FSTL1 was produced and purified by OBiO Technology. For FSTL1 expression experiment in vivo, a dose of $1.3 \times 10^{11}$ GC of AAV-FSTL1 was injected into the gastrocnemius muscle of MKO mice and a same dose of AAV-GFP was injected into the gastrocnemius muscle of MKO and control mice on 12-week NASH diet. The mice were sacrificed after 4 weeks of virus injection.

## Dual-luciferase reporter assays

The pGL3-FSTL1-promoter (−1525/+76) reporter plasmid was constructed by amplifying PCR products from C57BL/6J mouse genomic DNA and inserting it into the XhoI/HindIII site of the luciferase reporter vector pGL3-Vector (Promega) by using the following primers: GAACCTCGAGCTCTTAGCTTCAGGCTAT (F) and GAACAAGCTTAGG ACACTGGGCAGGGT (R). The pGL3-FSTL1-promoter (−1122/+76) reporter plasmid was constructed by amplifying PCR products from The pGL3-FSTL1-promoter (−1525/+76) reporter plasmid and inserting it into the XhoI/HindIIIsite of the luciferase reporter vector pGL3-Vector (Promega) by using the following primers: GAACCTCGAGCC ATGATACCTGGTATGATGAGAGC (F for −1122/+76) and GAACAA GCTTAGGACACTGGGCAGGGT (R for −1122/+76). The mutant reporter plasmids were constructed by amplifying the pGL3-FSTL1-promoter (−1525/+76) reporter plasmid by using the following primers: TGGTTAACTAGCACACCCCTCTTAAGTATAGAG (F for −1492/−1477 mutant reporter plasmid) and GGGTGTGCTAGTTAACCATGA TTGTTGAGTTGACATAGCCTGA (R for −1492/−1477 mutant reporter plasmid); ACATACCAACATCTGTGTAGCACTCGGGCTAT (F for −1254/−1232 mutant reporter plasmid) and CACAGATGTTGGTATGTA TGTAAACTCTGTTATGGGTCATATTGG (R for −1254/−1232 mutant reporter plasmid); TTGGTCTTCAGCTGTAGGTATTCAGCAGACCT (F for −1195/−1180 mutant reporter plasmid) and CTACAGCTGA AGACCAAAAAATTATGCCTAAATTCCACACTGA (R for −1195/−1180 mutant reporter plasmid). HEK293T cells were cultured in 24-well plates and co-transfected with pCDH-IRF4 plasmid, luciferase reporter plasmid, and pRL-TK (control reporter) by using ProFection Mammalian Transfection System (Promega) according to the manufacturer's instructions. After 48 h, cells were harvested and measured using the Dual-Luciferase Reporter assay system (Promega) according to the manufacturer's instructions. Luciferase activity was normalized to Renilla luciferase activity (control reporter).

## Chromatin immunoprecipitation (ChIP)-qPCR assay

We performed the ChIP assay using the Sonication chip Kit (ABclonal, China), according to the manufacturer's instructions. In brief, we fixed $2 \times 10^6$ C2C12 cells in 1% formaldehyde for 10 min at ambient temperature. The fixed cells were harvested, lysed, and sonicated for 66 cycles of 20 s ON/40 s OFF and 80% AMPL using Qsonica Q800R3 (Qsonica, USA). Antibodies against IRF4 (Cell Signaling Technology, USA) and mouse IgG (ABclonal, China) were used for immunoprecipitation. PCR amplification of the precipitated DNA was performed. The primer sequences were used for the ChIP assay as followed: ① forward primer, TCAAAACTAACTAGCACACCC, reverse primer, TACACAAGCAGGTTTTCCAA. ② forward primer, TTTTG TTTCCATCTGTGTAGC, reverse primer, CTTAGTTTATCGTCTACCG AG. ③ forward primer, TGAAACCTTCAGCTGTAGGT, reverse primer, CTTCTCTTCACCTTCTTTGC.

## Co-culture assay

To simulate the in vivo environment of NASH mice, HSC-T6 cells were induced fibrosis by 10 ng/ml TGF-β (MCE, China) factor[56]. AML12 cells and macrophage were induced lipid accumulation and inflammation by 200 nM palmitic acid (Sigma)[57]. The optimum concentration of rFSTL1 protein (MCE, China) was explored according to different cell lines. Packaging constructed shRNA of FSTL1 receptor into lentivirus. HSC-T6, AML12 and macrophage cells were first infected with lentivirus for 48 h, then induced with TGF-β or palmitic acid for 24 h, then co-cultured with gastrocnemius from Flox and MKO mice with/without rFSTL1 protein for 24 h. Finally, the cells were harvested for indicated experiments.

## Total RNA extraction, library construction, and sequencing

Three liver tissue samples of each group for RNA sequencing were collected. Total RNA was isolated using the Trizol method according to manufacturer's instructions. RNA quality was measured using the Agilent 2100 Bioanalyzer (RNA 6000 Nano Kit; Agilent Technologies, Santa Clara, CA, USA). cDNA libraries for each sample were constructed as reported previously. Libraries were sequenced on BGIseq500 platform (BGI-Shenzhen, China), using 150 bp paired-end reads aimed at 30 million reads per sample. The raw sequencing data was filtered with trimgalore (v0.6.7) by removing reads containing sequencing adapter and reads with low-quality base. The clean reads were mapped to the reference genome (mm10) using Hisat2 (2.2.1). Quantification of gene expression was calculated using FeatureCounts (v2.0.1), converted to transcripts per million (TPM) using the R software. We conducted a differential gene expression analysis comparing F4MKO versus control and overexpression of FSTL1 in F4MKO (F4MKO + FSTL1) versus F4MKO. Subsequently, differential expression analysis between groups was conducted using the DESeq2(1.3.40) with adjust P value < 0.05 and | FC| > 1.5. We used the obtained log2FC to perform a gene-set enrichment analysis to detect pathways enriched with profiling genes.

## Human serum proteomics

A total of 180 human serum samples (43 control, 43 NAFL, 41 eNASH, and 53 fNASH subjects) were used to perform proteomics analysis as previously described[58]. Human serum samples used for protein extraction were first removed the top 14 highest abundance proteins using an immunodepleting kit (Thermo Fisher) according to the manufacturer's instructions. The depleted serum was digested by trypsin at an enzyme to protein mass ratio of 1:25 overnight at 37 °C, and the peptides were then extracted and dried.

Samples were measured using LC-MS instrumentation consisting of an EASY- nLC 1200 ultra-high-pressure system (Thermo Fisher Scientific) coupled via a nano-electrospray ion source (Thermo Fisher Scientific) to a Fusion Lumos Orbitrap (Thermo Fisher Scientific). The peptides were dissolved with 12 μl loading buffer (0.1% formic acid in water), and 5 μl was loaded onto a 100 μm I.D. × 2.5 cm, C18 trap column at a maximum pressure 280 bar with 14 μl solvent A (0.1% formic acid in water). Peptides were separated on 150 μm I.D. × 15 cm column (C18, 1.9 lm, 120 #A, Dr. Maisch GmbH) with a linear 15–30% Mobile Phase B (ACN and 0.1% formic acid) at 600 nl/min for 75 min. The MS analysis was performed in a data-independent manner (DIA). The DIA method consisted of MS1 scan from 300–1400 $m/z$ at 60k resolution (AGC target 4e5 or 50 ms). Then, 30 DIA segments were acquired at 15k resolution with an AGC target 5e4 or 22 ms for maximal injection time.

The setting "inject ions for all available parallelizable time" was enabled. HCD fragmentation was set to normalized collision energy of 30%. The spectra were recorded in profile mode. The default charge state for the MS2 was set to 3.

The DIA data were searched against UniProt human protein database (updated on 2019.12.17, 20,406 entries) using FragPipe (v12.1) with MSFragger (2.2). The mass tolerances were 20 ppm for precursor and 50 mmu for product ions. Up to two missed cleavages were allowed. The search engine set cysteine carbamidomethylation as a fixed modification and N-acetylation and oxidation of methionine as variable modifications. Precursor ion score charges were limited to +2, +3, and +4. The data were also searched against a decoy database so that protein identifications were accepted at a false discovery rate (FDR) of 1%. The results of DIA data were combined into spectra libraries using SpectraST software. DIA data was analyzed using DIA-NN (v1.7.0). The default settings were used for DIA-NN (Precursor FDR: 5%, Log lev: 1, Mass accuracy: 20 ppm, MS1 accuracy: 10 ppm, Scan window: 30, Implicit protein group: genes, Quantification strategy: robust LC (high accuracy)). Quantification of identified peptides was calculated as the average of chromatographic fragment ion peak areas across all reference spectra libraries. Label-free protein quantifications were calculated using a label-free, intensity-based absolute quantification (iBAQ) approach. We calculated the peak area values as parts of corresponding proteins. The fraction of total (FOT) was used to represent the normalized abundance of a particular protein across samples. FOT was defined as a protein's iBAQ divided by the total iBAQ of all identified proteins within a sample. The FOT values were multiplied by $10^5$ for the ease of presentation and missing values were imputed with $10^{-5}$.

## Mouse serum proteomics

Mouse blood samples were obtained from the orbital vein and collected into commercial anticoagulation tubes containing tri-potassium ethylenediaminetetraacetic acid. Immediately after collection, the samples were centrifuged at $3000 \times g$ for 15 min at +4 °C to separate the serum. The supernatant was then frozen at −80 °C before LC-MS analysis.

To prepare the samples for LC-MS/MS analysis, the 14 most abundant serum proteins were removed from each sample using commercial depletion kits (High Select™ Depletion Spin Columns, Thermo Fisher Scientific), following the manufacturer's instructions. After the depletion process, the proteins were denatured, reduced, alkylated, digested into peptides, and desalted using the C-18 column. Then the samples were analyzed using LC-MS/MS equipment, which was conducted as previously described[59]. The MS spectra were acquired using a Data-Independent Acquisition (DIA) method. The DIA-MS method involved a preliminary MS1 scan covering the mass range of 300 to 1400 $m/z$ (with an AGC target of $4 \times 10^5$ and a maximum injection time of 50 ms) at a resolution of 60,000. Subsequently, 30 DIA segments were employed, each covering a specific mass range. These segments were scanned with an AGC target of $5 \times 10^4$ and a maximum injection time of 22 ms, at a resolution of 15,000. Peptide identification and protein quantification was performed according to the standard workflow in Skyline (https://skyline.ms/_webdav/home/software/Skyline/@files/tutorials). 6 raw files' reports were produced by Skyline DIA analysis and were merged to create an integrated expression matrix. This matrix includes the expression levels of each individual protein, utilizing all identified distinct peptides for protein quantification. The peptide and protein detection Q value was set at 5%. Proteome qualification was conducted using the iBAQ algorithm[60], and the resulting values were normalized to the fraction of the total (FOT). FOT is defined as the ratio of a protein's iBAQ to the total iBAQ of all identified proteins within a given sample, representing the normalized abundance of a specific protein across all samples. Any missing values were replaced with the 1/5 of minimal value.

## Quantification and statistical analysis

A two-tailed Student t test was performed for comparison of two groups. Two-way ANOVA followed by Bonferroni post-tests were performed for intergroup comparisons. All data except human clinical data were presented as mean ± standard error of mean (SEM).

Clinical data are shown as the mean ± standard deviation (SD). One-way ANOVA or the Kruskal–Wallis test was used for intergroup comparisons of continuous data, and the chi-squared test was used for comparisons of categorical variables. The Spearman's rank correlation coefficient was calculated for nonnormally distributed data.

## Reporting summary

Further information on research design is available in the Nature Portfolio Reporting Summary linked to this article.

## Data availability

Data supporting the findings of this study are available in the article and its Supplementary information. Source data are provided as Source data file. All data supporting the findings of this study are available in the Source data file. RNA-seq data has been deposited in the Gene Expression Omnibus (GEO) with accession codes: GSE216378. The proteomics data have been deposited in the PRIDE under accession number: PXD045324. Source data are provided with this paper.

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

## Acknowledgements

The authors gratefully acknowledge the platform of State Key Laboratory of Genetic Engineering at Fudan University. The authors gratefully acknowledge Analysis and Testing center of Experiment Center for Science and Technology, Shanghai University of Traditional Chinese Medicine and Human Phenome Institute, Fudan University for performing Proteomics. This work was funded by the National Natural Science Foundation of China (92157203) to X.K., the National Natural Science Foundation of China (32371194) to H.G., the National Key R&D Program of China (2019YFA0801900, 2018YFA0800300), the National Natural Science Foundation of China (31971074, 32150610475) to T.L., Science and Technology Commission of Shanghai Municipality (20ZR1410200) and Clinical Research Project of Zhongshan Hospital (2020ZSLC19) to H.B., the Yangfan Project of Shanghai Science and Technology Commission (grant number: 21YF1405000) and the National Natural Science Foundation of China (82300978) to Y.F., the National Natural Science Foundation of China (82100849) and Young Foundation of Zhongshan Hospital Fudan University (2021ZSQN07) to X.Z., the National Natural Science Foundation of China (82200952) to Q.Z.

## Author contributions

The experimental plan was designed by S.G., T.L. and X.K., S.G., Y.F., X.Zhu, H.G. and X.Zhang performed experiments and analyzed data. X.Zhang, Y.F. and H.W. collected the human samples and analyzed RNA-SEQ data. S.G., R.W. and Q.Z. carried out the animal experiments and analyzed the data. Y.R., Y.F., X.G., H.B. and X.Zhu collected human samples and analyzed the proteomics. X.K. wrote the manuscript incorporating edits and comments from T.L., X.G., Y.L., H.B. and all other authors.

## Competing interests

The authors declare no competing interests.
