## [Peer Review File · Nature Communications]

REVIEWER COMMENTS

Reviewer #1 (Remarks to the Author):

In the present study, Guo and colleagues investigated the role of muscle IRF4-FSTL1 in mediating NASH. They showed that IRF4 protein expression is increased in the muscle, but not the liver, of mice fed NASH diet for 24 weeks. Based on this observation, the authors generated muscle IRF4 knockout mice and showed that deleting muscle IRF4 reduces NASH in mice by suppressing muscle FSTL1. They further confirm this hypothesis by overexpressing FSTL1 in the muscle of IRF4 muscle knockout mice.

Specific Comments:

Although the concept is novel and the involvement of muscle IRF4-FSTL1 in NASH has yet to be shown before, it is hard to draw such a conclusion when the authors showed that FSTL1 mRNA level was significantly lower in liver biopsies from NASH patients. This point needs to be addressed by performing additional *in vivo* experiments.

Comments:

1. The authors proposed that deleting muscle IRF4 in mice fed NASH diet attenuates liver steatosis and inflammation by suppressing muscle FSTL1. However, the authors did not measure circulating FSTL1 in these mice nor in muscle IRF4 knockout mice with FSTL1 overexpression.
2. The authors need to show that mice lacking liver FSTL1 produce a similar phenotype as muscle IRF4 knockout mice (using adenovirus or *LoxP* approaches).
3. Assessment of glucose and energy homeostasis in muscle IRF4 knockout mice fed NASH diet would strengthen the mouse phenotype data.
4. In addition to assessing plasma liver's enzymes, ALT and AST, did the authors measure circulating lipids? Is it possible that reducing plasma lipids is responsible for the observed phenotype?
5. It is unclear whether the authors used male or female mice in the study. This needs to be clarified.

6. This manuscript has numerous typos, missing words, and irregular expressions. Thus, the language needs to be improved, and the discussion needs to be expanded.

Reviewer #2 (Remarks to the Author):

In this manuscript, Guo et al showed that skeletal muscle specific knockout IRF4(F4MKO) ameliorated liver steatosis, inflammation, and fibrosis under NASH diet. IRF4 transcriptionally regulates FSTL1 level, the gene encoding a myokine. Furthermore, reconstitution of FSTL1 expression in muscle of F4MKO mice inhibited the hepatic protective effects of F4MKO mice under a NASH diet. Co-culture experiments suggested that different receptors might contribute to FSTL1's function in different hepatic celltypes. Thus, a possible signaling pathway from skeletal muscle to liver via IRF4-FSTL1 in the pathogenesis of NASH was suggested. These authors have previously used F4MKO mice to study its effects on HFD and exercise, and in this study, they showed an organ crosstalk by a known myokine under NASH conditions. However, the majority of effects of FSTL1 on the liver were demonstrated by co-culture experiments. Here are specific comments for the authors.

1. It is interesting to know that muscle can affect liver pathology under NASH conditions. Are there any alterations in muscle phenotype, such as changes in muscle fiber size, number or types, of F4MKO mice in NASH conditions? The authors demonstrated that F4MKO mice induced downregulation of muscle FSTL1 under NASH conditions, did this also happen on NC conditions? Only gastrocnemius showed FSTL1 changes in F4MKO mice, how about other muscle parts?
2. Fig 1, the author need to provide both hepatic and serum triglyceride and cholesterol levels to better evaluate effects on F4MKO mice under NASH conditions.
3. Fig 2a, improved Oil red O staining results are needed (this also apply to other Oil red O staining in the manuscript); Fig 2g, compared with the flox group, the nuclei in the cells of F4MKO group were significantly larger. Please explain?
4. Fig 3d, statistical results are needed, and the alteration of FSTL1 in serum of F4MKO mice barely reach the statistical significance; the measurement of serum FSTL1 level using ELISA should provide. Furthermore, IRF4 knockout showed much greater effects on serum Col5a1, Decorin, Ldhd and Rbp4 levels, why Fstl1 was selected? Please further explain. Fig 3g, the quality of Western blot is poor.
5. Fig. 6, when one receptor of FSTL1 was knocked out, the expression levels of other receptors need to be evaluated. As shown in Extended Fig. 5, the FSTL1 receptors Dip2a and Cd14 are highly expressed in Kupffer cells, with very low expression level in HSC and hepatocytes. Immunostaining results on the livers under NASH conditions are needed to better identify which hepatic cell type is the mainly target

cell type for FSTL1. Furthermore, the authors need to provide TG content and lipid accumulation data in AML12 cells as they did in Fig 1g-h; while measure some inflammatory factors in macrophage.

6. More detail methods are needed, such as what kind of macrophage was used for the present study; whether human serum samples were used for proteomic studies? How rFSTL1 was obtained? etc.

7. Language needs improvement. Gene names should meet the NCBI requirements.

Reviewer #3 (Remarks to the Author):

In this work, Guo et al. investigated the role of the IRF4-FSTL1 signaling in skeletal muscle in NASH development. By using knockout mice and cell culture models, they show that skeletal muscle-specific IRF4 ablation protects against the development of NAFLD via the FSTL1-DIP2A/CD14 pathway. In addition, they show that serum FSTL1 level correlates with human NASH progression. Although the conclusion is largely supported by the data, some concerns should be addressed.

1) No data show that plasma FSTL1 levels are changed in IRF4 knockout mice or mice over-expressing IRF4. The data should be presented to support the conclusion.

2) A number of studies are performed in liver cell lines. The in vitro data may not be true in vivo. Does knockdown of the FSTL1 receptors DIP2A and/or CD14 prevent diet-induced NASH when FSTL1 is over-expressed in skeletal muscle?

3) Does ablation of FSTL1 in skeletal muscle prevent diet-induced NASH development?

4) What are the molecular mechanisms behind regulating lipogenic or fatty acid oxidation genes by the FSTL1-DIP2A/CD14 pathway?

5) The NASH diet contains 1.25% cholesterol which is supraphysiological.

6) The AAV-mediated over-expression of FSTL1 is not driven by a skeletal muscle-specific promoter. AAVs may also be taken up by the liver, which complicates the interpretation of the results.

7) NASH diet induces IRF4 in skeletal muscle. However, it is unclear whether FSTL1 protein level is also induced in skeletal muscle under the same condition.

8) Plasma FSTL1 levels have been shown to be elevated in liver steatosis and fibrosis (PMID: 30046009).

9) Does IRF4 bind to FSTL1 promoter in vivo?

10) It will be helpful to clearly state what kind of hepatocytes or macrophages (primary or cell lines) were used.

The following are our point-to-point responses to the comments and concerns raised by the reviewers:

REVIEWER COMMENTS

Reviewer #1 (Remarks to the Author):

In the present study, Guo and colleagues investigated the role of muscle IRF4-FSTL1 in mediating NASH. They showed that IRF4 protein expression is increased in the muscle, but not the liver, of mice fed NASH diet for 24 weeks. Based on this observation, the authors generated muscle IRF4 knockout mice and showed that deleting muscle IRF4 reduces NASH in mice by suppressing muscle FSTL1. They further confirm this hypothesis by overexpressing FSTL1 in the muscle of IRF4 muscle knockout mice.

Response: We highly thank reviewer #1 for his/her time and efforts in evaluating our manuscript. We appreciate the comments and concerns which help up further improve the quality of this study.

Specific Comments:

Although the concept is novel and the involvement of muscle IRF4-FSTL1 in NASH has yet to be shown before, it is hard to draw such a conclusion when the authors showed that FSTL1 mRNA level was significantly lower in liver biopsies from NASH patients. This point needs to address by performing additional in vivo experiments.

Response: We appreciate the Reviewer's insightful points. In our study, we found a significant decrease in both gastrocnemius FSTL1 protein expression (Fig 3a; Rebuttal Fig. 1a) and plasma FSTL1 levels (Fig 3c; Rebuttal Fig. 1b) in F4MKO mice, while we did not observe any significant alterations of FSTL1 protein expression (Extended Data Fig. 4e; Rebuttal Fig. 1c) in the liver. However, when we introduced skeletal muscle-specific overexpression (AAV2/9-MCS-FSTL1 or AAV-MCK promoter-FSTL1) of FSTL1 in F4MKO mice, we were able to rescue the muscle FSTL1 expression (Fig 4a; Rebuttal Fig. 1d) and plasma FSTL1 levels (Fig 4c; Rebuttal Fig. 1e). Additionally, we observed a significant increase in liver FSTL1 protein expression (Extended Data Fig. 4e; Rebuttal Fig. 1c) after AAV injection, but there was no obvious change in liver

Fstl1 mRNA expression (Extended Data Fig. 4f; Rebuttal Fig. 1f). These findings suggested that skeletal muscle plays a critical role in maintaining normal levels of circulating FSTL1.

Furthermore, we performed knockdown of FSTL1 either in the liver or in the gastrocnemius by using AAV methods. The results showed that knockdown of FSTL1 in the liver did not have any significant effects on NASH liver phenotypes (Extended Data Fig. 5g-k; Rebuttal Fig. 2a-f). These results are now described in the text (page 8, lines 191-193). However, knockdown of FSTL1 in the gastrocnemius led to a mild decrease in liver fibrosis of NASH mice assessed by sirius red staining (Extended Data Fig. 5b; Rebuttal Fig. 3a-f). These results are now described in the text (page 8, lines 189-190). Concomitant with this, fibrosis genes (*Acta2*, *Col1a1*) were decreased (Extended Data Fig. 5f; Rebuttal Fig. 3g), indicating that knockdown of FSTL1 in the gastrocnemius could ameliorate the liver fibrosis phenotype, which was consistent with the phenotype observed in F4MKO mice. These results are now described in the text (page 8, lines 191-193). Additionally, gastrocnemius-specific overexpression of FSTL1 (AAV-MCK promoter-FSTL1) aggravated liver steatosis and fibrosis (Rebuttal Fig. 4a-f). These in vivo studies were consistent with previous studies showed that skeletal muscle, a major source of FSTL1 production, is capable of secreting physiologically relevant levels of this factor¹. These results are now described in the text. We also added more discussion about the above results, which described in the text (page 13, lines 321-325).

In addition, in our study, the human data showed the mRNA level of *Fstl1* was decreased in NASH liver compared with healthy normal-weight individuals (Fig 7i; Rebuttal Fig. 5b), which was consistent with Suppli's study². Nonetheless, we observed that the serum FSTL1 level was elevated in NASH patients (Fig 7d; Rebuttal Fig. 5a). Though it will be more informative to measure the FSTL1 levels in muscle from human subjects (it is hard to collect both muscle and liver biopsies from human subjects), given others' and our studies, the increased plasma FSTL1 in NASH patients might be secreted from muscle.

Rebuttal Fig. 2

(a) Relative mRNA expression of *Fstl1* in the liver of NASH+GFP, NASH+LFSTL1-KD mice (n=4). (b-d) Plasma TG, ALT, and AST in NASH+GFP, NASH+LFSTL1-KD mice (n=4). (e and f) H&E and Sirius red staining of liver in NASH+GFP, NASH+LFSTL1-KD mice (Scale bars, 50 μ m). All results were shown as mean \pm SEM. **** p <0.0001, compared with NASH+GFP group.

Rebuttal Fig. 3

(a) Relative mRNA expression of *Fstl1* in the GAS of NASH+GFP, NASH+GFSTL1-KD mice (n=4). (b-d) Plasma TG, ALT, and AST in NASH+GFP, NASH+GFSTL1-KD mice (n=4). (e and f) H&E and Sirius red staining of liver in NASH+GFP, NASH+GFSTL1-KD mice (Scale bars, 50 μ m). (g) Relative mRNA expression of lipid metabolism related genes, fibrosis genes, and inflammatory genes in NASH liver from NASH+GFP, NASH+GFSTL1-KD mice (n=4). All results were shown as mean \pm SEM. * p <0.05, ** p <0.01, **** p <0.0001, compared with NASH+GFP group.

Comments:

1. The authors proposed that deleting muscle IRF4 in mice fed NASH diet attenuates liver steatosis and inflammation by suppressing muscle FSTL1. However, the authors did not measure circulating FSTL1 in these mice nor in

muscle IRF4 knockout mice with FSTL1 overexpression.

Response: As suggested, during the revision, we have performed ELISA experiments to determine the circulating FSTL1 of the F4MKO and control mice fed with NASH diet for 24 weeks. The results showed that the circulating FSTL1 was significantly decreased in F4MKO mice compared with control mice group (Fig 3c; Rebuttal Fig. 1b). These results are now described in the text (page 6, lines 150-152). However, the circulating FSTL1 levels were rescued after FSTL1 overexpression (Fig 4b, c; Rebuttal Fig. 1e, 6). These results are now described in the text (page 7, lines 174-175).

2. The authors need to show that mice lacking liver FSTL1 produce a similar phenotype as muscle IRF4 knockout mice (using adenovirus or LoxP approaches).

Response: As suggested, during the revision we performed knockdown of FSTL1 in liver by using AAV method. The results showed that knockdown of FSTL1 had no significant effects on liver metabolism (Extended Data Fig. 5g-k, Rebuttal Fig. 2a-f). These results are now described in the text (page 8, lines 189-190). Such results might firstly be due to the fact that FSTL1 was only knocked down in the liver but not in other tissues, especially in the muscle. The circulating FSTL1 was not significantly different in KD and control mice (Rebuttal Fig. 7a). Secondly, in Suppli's² and our studies, the NASH patients with lower Fstl1 expression in liver but higher levels in plasma, but the liver Fstl1 expression was not altered in F4MKO mice (ameliorated NASH) compared with controls (Rebuttal Fig. 1f), suggesting the expression level of liver derived-Fstl1 was not associated with NASH progression.

Meanwhile, we performed experiments by using AAV-shFSTL1 to knockdown FSTL1 in muscle to observe the effects on NASH. The results showed that knockdown of FSTL1 in skeletal muscle had partially alleviated NASH phenotypes (Extended Data Fig. 5a-f; Rebuttal Fig. 3a-g), and the circulating FSTL1 was decreased in muscle KD mice (Rebuttal Fig. 7b), indicating that muscle-derived FSTL1 plays an essential role in the development of NASH. These results were described in the text (page8, lines 186-191).

3. Assessment of glucose and energy homeostasis in muscle IRF4 knockout mice fed NASH diet would strengthen the mouse phenotype data.

Response: As suggested, during the revision, we performed metabolic cage experiments to assess the basal metabolism and energy expenditure of the F4MKO and control mice fed with NASH diet for 24 weeks. The results showed that the oxygen consumption rate and carbon dioxide production rate were not significantly altered in KO mice compared to those in control mice (Extended Data Fig. 1l, 1m; Rebuttal Fig. 8a, 8b). These results are now described in the text (page 4, lines 95-97).

Similarly, the respiratory exchange ratio (RER) and energy expenditure (EE) were not significantly different in KO and control mice (Extended Data Fig. 1n, 1o; Rebuttal Fig. 8c, 8d). These results are now described in the text (page 4, lines 95-97).

Furthermore, in our previous study, we performed GTT and ITT experiments, the results showed that knockout IRF4 in muscle ameliorated glucose intolerance and insulin sensitivity³.

4. In addition to assessing plasma liver's enzymes, ALT and AST, did the authors measure circulating lipids? Is it possible that reducing plasma lipids is responsible for the observed phenotype?

Response: As suggested, the blood levels of TG and TC in mice from the F4MKO mice and control mice were measured. The results showed that the levels of plasma TG and TC were slightly, but not significantly decreased in F4MKO mice fed on NASH diet for 24 weeks (Extended Data Fig. 2c, 2d; Rebuttal Fig. 9a, 9b). These results are now described in the text (page 5, lines 109-110).

Rebuttal Fig. 9

(a and b) Plasma TG and TC in male Flox and F4MKO mice on NASH diet (n=6). All results were shown as mean \pm SEM.

5. It is unclear whether the authors used male or female mice in the study. This needs to be clarified.

Response: We are sorry for not providing enough experimental details, in the study we investigated both male and female mice to avoid the gender differences. We added the female results to the manuscript (Extended data Fig 1r-t; Rebuttal Fig. 10a-c). These results were described in the text (page 4, lines 102-103). More details were provided for Materials and Methods (page 19, lines 490).

Rebuttal Fig. 10

(a) H&E staining of liver in female Flox and F4MKO mice (Scale bars, 50 μ m). (b and c) Serum ALT and AST levels in female Flox and F4MKO mice on 24-weeks NASH diet (n=5). All results were shown as mean \pm SEM. ** p <0.01, *** p <0.001, compared with the female Flox group.

6. This manuscript has numerous typos, missing words, and irregular expressions. Thus, the language needs to be improved, and the discussion needs to be expanded.

Response: We reviewed and revised the manuscript to improve the English. KetengEdit (www.ketengedit.com) offered linguistic assistance for manuscript and certified, as we provided in the attached file.

The discussion section has been modified as suggested and now described in the text (page 13, lines 313-329). Thank you!

Reviewer #2 (Remarks to the Author):

In this manuscript, Guo et al showed that skeletal muscle specific knockout IRF4(F4MKO) ameliorated liver steatosis, inflammation, and fibrosis under NASH diet. IRF4 transcriptionally regulates FSTL1 level, the gene encoding a myokine. Furthermore, reconstitution of FSTL1 expression in muscle of F4MKO mice inhibited the hepatic protective effects of F4MKO mice under a NASH diet. Co-culture experiments suggested that different receptors might contribute to FSTL1's function in different hepatic cell types. Thus, a possible signaling pathway from skeletal muscle to liver via IRF4-FSTL1 in the pathogenesis of NASH was suggested. These authors have previously used F4MKO mice to study its effects on HFD and exercise, and in this study, they showed an organ crosstalk by a known myokine under NASH conditions. However, the majority of effects of FSTL1 on the liver were demonstrated by co-culture experiments. Here are specific comments for the authors.

Response: We greatly appreciate the time and efforts from this expert reviewer to evaluate our manuscript. His/her considerate and constructive suggestions largely help us to further improve the quality of this study.

1. It is interesting to know that muscle can affect liver pathology under NASH conditions. Are there any alterations in muscle phenotype, such as changes in muscle fiber size, number or types, of F4MKO mice in NASH conditions? The authors demonstrated that F4MKO mice induced downregulation of muscle FSTL1 under NASH conditions, did this also happen on NC conditions? Only gastrocnemius showed FSTL1 changes in F4MKO mice, how about other muscle parts?

Response: As suggested, during the revision, we performed experiments to determine the muscle phenotype. The results showed that knockout of IRF4 in muscle had no effects on muscle fiber size, number or fiber types under NASH condition (Extended data Fig. 1p, 1q; Rebuttal Fig. 11a, 11b). These results were described in the text (page 4, lines 95-97).

We performed experiments to determine the muscle FSTL1 expression under normal chow diet (NCD) condition in control mice and F4MKO mice. The results showed that there were no differences between F4MKO mice and control mice under NCD condition (Extended data Fig. 4a; Rebuttal Fig. 11c). These results were described in the text (page 6, lines 148-151).

As suggested, we investigated the expression of FSTL1 in different muscle in control mice and F4MKO mice. The FSTL1 expression was also decreased in soleus and quadriceps muscle in F4MKO mice compared with control mice (Extended data Fig. 4b, 4c; Rebuttal Fig. 11d, 11e). These results were described in the text (page 6, lines 148-151).

2. Fig 1, the author need to provide both hepatic and serum triglyceride and cholesterol levels) to better evaluate effects on F4MKO mice under NASH conditions.

Response: Thanks for the suggestions. We performed the hepatic TG and TC in the Flox and F4MKO mice fed on NASH diet for 24 weeks. The results showed that hepatic TG and TC were both decreased in F4MKO mice under NASH condition (Extended Data Fig. 2a, 2b; Rebuttal Fig. 12a, 12b). These results were described in the text (page 5, lines 107-109).

As suggested, the plasma of TG and TCH level in mice from the F4MKO mice and Flox mice were also measured. The results showed that the level of plasma TG and TC were slightly, but not significantly, decreased in F4MKO mice fed on NASH diet for 24 weeks (Extended Data Fig. 2c, 2d; Rebuttal Fig. 9a, 9b). These results are now described in the text (page 5, lines 109-110).

3. Fig 2a, improved Oil red O staining results are needed (this also apply to other Oil red O staining in the manuscript); Fig 2g, compared with the flox group, the nuclei in the cells of F4MKO group were significantly larger. Please explain
Response: The Oil red O staining images with better quality are now provided (Fig 2a, Fig 5c; Rebuttal Fig. 13a, 13b). And the liver Oil red O staining images of F4MKO+FSTL1 mice were replaced with AAV-MCK promoter FSTL1 (Fig 5c; Rebuttal Fig. 13b). These results were described in the text (page 5, lines 107-108; page 8, lines 204-205).

The larger nuclei in the cells of F4MKO group probably due to we used primary hepatocyte. Hepatocyte ploidy is one of the characteristics of primary hepatocytes, and some hepatocytes have larger polyploid nuclei^{4, 5}. We replaced with new data.

4. Fig 3d, statistical results are needed, and the alteration of FSTL1 in serum of F4MKO mice barely reach the statistical significance; the measurement of serum FSTL1 level using ELISA should provide. Furthermore, IRF4 knockout showed much greater effects on serum Col5a1, Decorin, Ldhd and Rbp4 levels, why Fstl1 was selected? Please further explain. Fig 3g, the quality of Western blot is poor.

Response: As suggested, during the revision, we performed ELISA experiments to determine the plasma FSTL1 level of the control mice and

F4MKO mice under NASH condition (Fig. 3c; Rebuttal Fig. 1b). These results were described in the text (page 6, lines 151-152).

Serum proteomics from F4MKO and Flox mice identified 376 differentially expressed proteins corresponding to fold change >1.5 or fold change <0.67; 194 up-regulated proteins, 182 down-regulated proteins. To obtain the differently expressed (DE) proteins derived from skeletal muscle, we overlapped DE proteins with DE genes from transcriptome sequencing data between F4MKO muscle and IRF4 specific-muscle overexpression (F4MOE) 18 and found 15 down-regulated and 3 up-regulated proteins. Next, we screened the 18 proteins with reported potential myokines, 12 down-regulated and 1 up-regulated proteins belonged to myokines. To further validate changes in myokines secreted by skeletal muscle, we examined genes coding these DE proteins in gastrocnemius of F4MKO and Flox NASH mouse models. We found that Decorin, Fibronectin1, Fstl1 and Mmp2 were significantly decreased in gastrocnemius of F4MKO mice (Extended Data Fig. 3e, Rebuttal Fig. 14a). Fstl1, Mmp2 and Decorin were reported to be associated with NASH. However, Mmp2⁶ and Decorin^{7, 8} provide protective role on NASH, down-regulation of them will worsen liver pathology. Fstl1 was reported to promote fibrosis in liver⁹. We thus chose Fstl1 as a candidate to link muscle and liver in our models.

WB images with better quality are now provided (Fig. 3b; Rebuttal Fig. 14b).

5. Fig. 6, when one receptor of FSTL1 was knocked out, the expression levels of other receptors need to be evaluated. As shown in Extended Fig. 5, the FSTL1 receptors Dip2a and Cd14 are highly expressed in Kupffer cells, with very low expression level in HSC and hepatocytes. Immunostaining results on the livers under NASH conditions are needed to better identify which hepatic cell type is the mainly target cell type for FSTL1. Furthermore, the authors need to provide TG content and lipid accumulation data in AML12 cells as they did in Fig 1g-h; while measure some inflammatory factors in macrophage.

Response: Thanks for your suggestion. The expression levels of other receptors were evaluated (Extended Data Fig. 8h, 8i, 9h, 10f; Rebuttal Fig. 15 a-d). These results were described in the text (page 10, lines 239-240, 243-244, 248-249).

Although the expression of FSTL1 receptors Dip2a and Cd14 is very low in HSCs and hepatocytes. In our study, co-culture experiments showed that rFSTL1 increased lipid accumulation in AML12 cells, but decreased in DIP2A/CD14 blocked cells (Fig 6a-h; Rebuttal Fig. 16a-h). The fibrosis genes were increased in rFSTL1 supplement cells but decreased only in shDip2a infected cells (Fig 6n-r; Rebuttal Fig. 16i-m). So FSTL1 also acts on hepatocytes and macrophages.

And in vivo experiments have also shown both Dip2a and Cd14 knockdown in the liver, FSTL1 could not rescued hepatic damage, lipid accumulation and fibrosis (Figure 6s-w; Rebuttal Fig. 17a-g). These results were described in the text (page 10, lines 257-260).

TG content and lipid accumulation data in AML12 cells were provided (Extended Data Fig. 2l, 2m; Rebuttal Fig. 18a, 18b). These results were described in the text (page 5, lines 127).

Rebuttal Fig. 16

(a-h) Lipid metabolism genes of AML12 cells co-cultured with GAS from Flox or F4MKO mice. AML12 cells co-cultured with GAS from Flox or F4MKO mice, were infected with shDip2a or shCd14 lentivirus and then treated with 200 μ M PA for 24h. In shScramble+rFSTL1 and shDip2a+rFSTL1 or shCd14+rFSTL1 group, AML12 cells were additionally treated with 100 ng/ml rFSTL1. (i-m) Relative mRNA expression of fibrosis genes in HSC-T6 cells co-cultured with GAS from Flox and F4MKO mice. HSC-T6 cells co-cultured with GAS from Flox and F4MKO mice, were infected with shDip2a lentivirus and then treated with 10ng/ml TGF β 1 for 24h. In shscramble+rFSTL1 and shDip2a+rFSTL1 group, HSC-T6 cells were additionally treated with 100 ng/ml rFSTL1. All results were shown as mean \pm SEM.

Rebuttal Fig. 18

(a) Immunofluorescence of AML12 cells co-cultured with gastrocnemius from Flox and F4MKO mice. AML12 cells treated with 200 μ M PA were co-cultured with gastrocnemius from Flox and F4MKO mice for 24 h. (b) Cellular TG level of AML12 cells from a. All results were shown as mean \pm SEM, **** p <0.0001. Compared with the Flox group.

6. More detail methods are needed, such as what kind of macrophage was used for the present study; whether human serum samples were used for proteomic studies? How rFSTL1 was obtained? etc.

Response: We are sorry for not providing enough experimental details.

The mouse RAW264.7 macrophage cell line was used for the study (page 21, lines 544-546). Human serum samples were used for proteomic studies (page 25, lines 641). The rFSTL1 was obtained from MedChemExpress (MCE) (page 24, lines 619-620). More details were provided for experiments and Materials and Methods in the revised version.

7. Language needs improvement. Gene names should meet the NCB I requirements.

Response: We reviewed and revised the manuscript to improve the English. KetengEdit (www.ketengedit.com) offered linguistic assistance for manuscript and certified, as we provided in the attached file.

Reviewer #3 (Remarks to the Author):

In this work, Guo et al. investigated the role of the IRF4-FSTL1 signaling in skeletal muscle in NASH development. By using knockout mice and cell culture models, they show that skeletal muscle-specific IRF4 ablation protects against the development of NAFLD via the FSTL1-DIP2A/CD14 pathway. In addition, they show that serum FSTL1 level correlates with human NASH progression. Although the conclusion is largely supported by the data, some concerns should be addressed.

Response: We highly thank reviewer #3 for his/her helpful comments, concerns and suggestions, which greatly help us revise this manuscript. Please see below our point-to-point responses to the points.

1) No data show that plasma FSTL1 levels are changed in IRF4 knockout mice or mice over-expressing IRF4. The data should be presented to support the conclusion.

Response: As suggested, during the revision, we performed ELISA experiments to determine the circulating FSTL1 of the F4MKO and Flox mice fed with NASH diet for 24 weeks. The results showed that the circulating FSTL1 was significantly decreased in F4MKO mice compared with control mice (Fig 3c; Rebuttal Fig. 1b). As requested, these results were described in the text (page 6, lines 151-152).

2) A number of studies are performed in liver cell lines. The in vitro data may not be true in vivo. Does knockdown of the FSTL1 receptors DIP2A and/or

CD14 prevent diet-induced NASH when FSTL1 is over-expressed in skeletal muscle?

Response: As suggested, during the revision, we constructed the adeno-associated virus of DIP2A and CD14. We knocked down the DIP2A and CD14 receptors in the liver when FSTL1 was over-expressed in the skeletal muscle. We found that both knockdown of DIP2A and CD14 receptors in the liver decreased plasma ALT and AST (Fig 6s, 6t, Rebuttal Fig. 17c, 17d), reduced the liver lipid accumulation (Fig 6u-v; Rebuttal Fig. 17 e-f) and fibrosis (Fig 6w; Rebuttal Fig. 17g) under NASH diet challenge. These findings indicated that knockdown of the FSTL1 receptors DIP2A and CD14 prevent diet-induced NASH when FSTL1 was over-expressed in skeletal muscle. These results were described in the text (page 10, lines 257-260).

Rebuttal Fig. 17

(a and b) Relative mRNA expression of Dip2a and Cd14 in the liver of Flox+GFP and F4MKO+FSTL1+DC KD (Dip2a and Cd14 knock down) mice (c and d), Plasma ALT and AST levels in Flox+GFP, F4MKO+GFP, F4MKO+FSTL1, and F4MKO+FSTL1+DC KD mice on 24-weeks NASH diet. (e) Liver TG level in Flox+GFP, F4MKO+GFP, F4MKO+FSTL1, and F4MKO+FSTL1+DC KD mice. (f and g), H&E staining and Sirius red staining of NASH liver in Flox+GFP, F4MKO+GFP, F4MKO+FSTL1, and F4MKO+FSTL1+DC KD mice (scale bars, 50 μ m). All results are shown as means \pm SEM. ** p <0.01, *** p <0.001, **** p <0.0001, compared with the Flox+GFP group; ### p <0.001, compared with the F4MKO group; \$ p <0.05, \$\$ p <0.01, compared with the F4MKO+FSTL1 group.

3) Does ablation of FSTL1 in skeletal muscle prevent diet-induced NASH development?

Response: Thank you for this constructive suggestion! As suggested, we performed experiments by using AAV-shFSTL1 to knockdown FSTL1 in muscle to observe the effects on NASH. The results showed that knockdown of FSTL1 in skeletal muscle had partially alleviated NASH phenotypes (Extended Data Fig. 5a-f; Rebuttal Fig. 3a-g), which indicated that muscle-derived FSTL1 plays an essential role in the development of NASH. These results were described in the text (page 8, lines 186-191).

Additionally, we also observed a mild aggravated liver steatosis and fibrosis after using AAV (AAV-MCK promoter-FSTL1) to specific overexpress FSTL1 in gastrocnemius under NASH diet (Rebuttal Fig. 4a-f).

expression of lipid metabolism related genes, fibrosis genes, and inflammatory genes in NASH liver from NASH+GFP, NASH+GFSTL1-KD mice (n=4). All results were shown as mean \pm SEM. * p <0.05, ** p <0.01, **** p <0.0001, compared with NASH+GFP group.

4) What are the molecular mechanisms behind regulating lipogenic or fatty acid oxidation genes by the FSTL1-DIP2A/CD14 pathway?

Response: Previous studies identified that FSTL1 as a secreted protein that can bind to CD14 or DIP2A, which act as membrane receptors^{10, 11}.

In cardiomyocytes, after being activated by FSTL1, DIP2A enhances the activation of the downstream Akt¹². FSTL1 promoted proliferation, migration, and invasion in gastric cancer, at least partially, by activating AKT via regulating TLR4/CD14 in gastric tissue¹³. Activation of Akt has been shown to increase hepatic lipogenesis^{14, 15}, this might be a potential molecular mechanism by which FSTL1-DIP2A/CD14 pathway regulates lipogenesis genes.

5) The NASH diet contains 1.25% cholesterol which is supraphysiological.

Response: We agreed that the choice of experimental models is important and must be well-justified. Animal models of NASH are typically based on different diets, such as high-fat, high-glucose, sucrose, fructose, methionine and choline-deficient, choline-deficient L-amino-defined, high-cholesterol, and cholesterol and cholate diets. It has been observed that some WDs contain extremely high levels of cholesterol (1-2%), which is significantly higher than the recommended daily intake for humans and may not be feasible to consume. Nevertheless, it is important to note that in order to accurately model human NASH, higher levels of dietary cholesterol may be necessary in rodents, especially in the commonly used C57BL6/J mice, due to inherent differences between mice and humans¹⁶.

In this study, increasing cholesterol (~1.25%) accelerates the NASH phenotype with steatosis, inflammation and hepatocyte ballooning as previously described^{17, 18}. In addition to feeding a high fat diet, providing a glucose/fructose mixture in the drinking water may further promote NASH development.

6) The AAV-mediated over-expression of FSTL1 is not driven by a skeletal muscle-specific promoter. AAVs may also be taken up by the liver, which complicates the interpretation of the results.

Response: As suggested, we performed new experiments by using AAV-MCK promoter-FSTL1 to specifically overexpress FSTL in muscle to investigate the effects on NASH. The results showed that FSTL1 mRNA expression was specifically increased in muscle but not liver, demonstrating that AAV targeting specific to muscle (Extended Data Fig. 4d, 4f; Rebuttal Fig. 20a, 1f). These results were described in the text (page 7, lines 171-173, lines 176-178).

The results demonstrated that while injected with AAV-MCK promoter FSTL1, overexpression of FSTL1 could counteract the effects of skeletal muscle- IRF4-ablation induced steatosis, fibrosis, and inflammation of NASH mice (Fig 4d-e, Fig 5c-h; Rebuttal Fig. 19a-b, Rebuttal Fig.13b, Rebuttal Fig. 19e-i). These findings were consistent with previous experimental results obtained from injection of AAV-FSTL1. We replaced data pertaining to F4MKO+FSTL1 in the manuscript with results obtained from the injection of AAV-MCK promoter-FSTL1 (Fig 4a-c, 4f-g, Extended Data 4e, 4g-k, Extended Data 6a, Extended Data 7e; Rebuttal Fig. 1c, Rebuttal Fig. 6, Rebuttal Fig. 1e, Rebuttal Fig. 19c-d, Rebuttal Fig. 1d, Rebuttal Fig. 20b-h).

Rebuttal Fig. 1f

Rebuttal Fig. 13b

Rebuttal Fig. 1c

Rebuttal Fig. 1e

Rebuttal Fig. 6

Rebuttal Fig. 1d

(Rebuttal Fig. 1f) Relative mRNA expression of *Fstl1* in the liver (n=6). (Rebuttal Fig. 13b) Oil red O staining of liver in Flox+GFP, F4MKO+GFP, and F4MKO+FSTL1 mice (scale bars, 50 μm). (Rebuttal Fig. 1c) Western blot analysis of the expression of FSTL1 and FLAG in skeletal muscle of Flox+GFP, F4MKO+GFP and F4MKO+FSTL1 mice. (Rebuttal Fig. 6) Western blot analysis of the expression of FSTL1 and FLAG in plasma of Flox+GFP, F4MKO+GFP and F4MKO+FSTL1 mice. (Rebuttal Fig. 1e) Elisa analysis of the level of plasma FSTL1 from mice of Rebuttal Fig. 1c. (Rebuttal Fig. 1d) Western blot analysis of the expression of FSTL1 and FLAG in skeletal muscle of Flox+GFP, F4MKO+GFP and F4MKO+FSTL1 mice.

7) NASH diet induces IRF4 in skeletal muscle. However, it is unclear whether FSTL1 protein level is also induced in skeletal muscle under the same condition. Response: Thanks. We performed the experiment to determine the FSTL1 expression under NASH condition. The results showed that under NASH condition, FSTL1 expression was significantly increased (Figure 3b, Rebuttal Fig. 14). These results were described in the text (page 6, lines 152-153).

Rebuttal Fig. 14

Western blot analysis of the expression of FSTL1 in skeletal muscle of CON, NASH and NASH+Exe mice.

8) Plasma FSTL1 levels have been shown to be elevated in liver steatosis and fibrosis (PMID: 30046009).

Response: We agreed with reviewer's opinion, and cited this paper in our manuscript (page 3, lines 56-57).

9) Does IRF4 bind to FSTL1 promoter in vivo?

Response: As suggested, during the revision, we performed ChIP assay to determine whether IRF4 binds to FSTL1 promoter. As expected, our results showed that IRF4 binds to FSTL1 promoter (Figure 3f; Rebuttal Fig. 21). These results are now described in the text (page 7, lines 166-167).

Rebuttal Fig. 21

QPCR analysis of each ChIP-DNA sample was performed for FSTL1, β -ACTIN. Results are reported as fold enrichment of immunoprecipitated DNA from each sample relative to the DNA immunoprecipitated with the non-specific antibody, and were plotted in a scale in which the final value of IgG was arbitrarily set to 1.

10) It will be helpful to clearly state what kind of hepatocytes or macrophages (primary or cell lines) were used.

Response: More details were provided for experiments and Materials and Methods (page 21, lines 544-548).

References

1. Miyabe M, *et al.* Muscle-derived follistatin-like 1 functions to reduce neointimal formation after vascular injury. *Cardiovasc Res* **103**, 111-120 (2014).
2. Suppli MP, *et al.* Hepatic transcriptome signatures in patients with varying degrees of nonalcoholic fatty liver disease compared with healthy normal-weight individuals. *Am J Physiol Gastrointest Liver Physiol* **316**, G462-g472 (2019).
3. Yao T, *et al.* Obese Skeletal Muscle-Expressed Interferon Regulatory Factor 4 Transcriptionally Regulates Mitochondrial Branched-Chain Aminotransferase Reprogramming Metabolome. *Diabetes* **71**, 2256-2271 (2022).
4. Gentric G, Desdouets C. Polyploidization in liver tissue. *Am J Pathol* **184**, 322-331 (2014).
5. Donne R, Saroul-Ainama M, Cordier P, Celton-Morizur S, Desdouets C. Polyploidy in liver development, homeostasis and disease. *Nat Rev Gastroenterol Hepatol* **17**, 391-405 (2020).
6. Roeb E. Matrix metalloproteinases and liver fibrosis (translational aspects). *Matrix Biol* **68-69**, 463-473 (2018).
7. Baghy K, *et al.* Ablation of the decorin gene enhances experimental hepatic fibrosis and impairs hepatic healing in mice. *Lab Invest* **91**, 439-451 (2011).
8. Ma R, He S, Liang X, Yu H, Liang Y, Cai X. Decorin prevents the development of CCl₄-induced liver fibrosis in mice. *Chin Med J (Engl)* **127**, 1100-1104 (2014).
9. Xu XY, *et al.* Targeting Follistatin like 1 ameliorates liver fibrosis induced by carbon tetrachloride through TGF- β 1-miR29a in mice. *Cell Commun Signal* **18**, 151 (2020).
10. Murakami K, *et al.* Follistatin-related protein/follistatin-like 1 evokes an innate immune response via CD14 and toll-like receptor 4. *FEBS Lett* **586**, 319-324 (2012).
11. Ouchi N, *et al.* DIP2A functions as a FSTL1 receptor. *J Biol Chem* **285**, 7127-7134 (2010).
12. Xi Y, Hao M, Liang Q, Li Y, Gong DW, Tian Z. Dynamic resistance exercise increases skeletal muscle-derived FSTL1 inducing cardiac angiogenesis via DIP2A-Smad2/3 in rats following myocardial infarction. *J Sport Health Sci* **10**, 594-603 (2021).

13. Wu M, *et al.* FSTL1 promotes growth and metastasis in gastric cancer by activating AKT related pathway and predicts poor survival. *Am J Cancer Res* **11**, 712-728 (2021).
14. Calvisi DF, *et al.* Increased lipogenesis, induced by AKT-mTORC1-RPS6 signaling, promotes development of human hepatocellular carcinoma. *Gastroenterology* **140**, 1071-1083 (2011).
15. Hagiwara A, *et al.* Hepatic mTORC2 activates glycolysis and lipogenesis through Akt, glucokinase, and SREBP1c. *Cell Metab* **15**, 725-738 (2012).
16. Gallage S, *et al.* A researcher's guide to preclinical mouse NASH models. *Nat Metab* **4**, 1632-1649 (2022).
17. Zhang K, *et al.* Deficiency of the Mitochondrial NAD Kinase Causes Stress-Induced Hepatic Steatosis in Mice. *Gastroenterology* **154**, 224-237 (2018).
18. Ichimura M, *et al.* A diet-induced Sprague-Dawley rat model of nonalcoholic steatohepatitis-related cirrhosis. *J Nutr Biochem* **40**, 62-69 (2017).

REVIEWER COMMENTS

Reviewer #1 (Remarks to the Author):

I am pleased to say that the manuscript has significantly improved and has addressed my main concerns. Therefore, I fully support its publication without any further experiments. However, there are still a few issues that could be resolved by addressing the following points as caveats or limitations.

1. Reviewer three's point regarding the high cholesterol content (1.25%) in the NASH diet, compared to the typical cholesterol levels (0.21%) reported in the diet, is valid. As a result, it would be appropriate to include this as a limitation in the study and provide additional information on the macro- and micro-nutrient composition of the diet.

2. The authors used an fl/fl mouse as a control, but previous research (PMID: 29858650) has shown that activating Cre recombinase in skeletal muscle can improve glucose tolerance. Therefore, when investigating skeletal muscle gene/protein function in the context of obesity using Cre-LoxP systems, it's important to include a Cre mouse as a necessary control littermate. This should be taken into consideration when planning future studies.

I would like to extend my congratulations to the authors for producing such a well-crafted paper.

Reviewer #2 (Remarks to the Author):

Authors addressed the questions raised by the reviewer.

Reviewer #3 (Remarks to the Author):

Although the authors addressed some of the concerns, some other concerns remain in place:

- 1) Plasma FSTL1 levels in mice overexpressing IRF4 are not shown. The revised manuscript only presents plasma FSTL1 levels in mice lacking IRF4.
- 2) The new data (Rebuttal Fig. 17) show that the knockdown of DIP2A and CD14 receptors reduces liver steatosis and injury when FSTL1 is overexpressed in skeletal muscle, suggesting that FSTL1 regulates the development of NASH independent of the two receptors. Thus, how FSTL1 interacts with hepatocytes to regulate NASH development remains unclear.
- 3) AAV-shFSTL1 can cause the knockdown of FSTL1 in other cell types as well. It does not have a muscle-specific promoter to drive the FSTL1 knockdown. Importantly, the knockdown of GAS FSTL1 does not affect plasma ALT or AST levels or liver phenotype (Rebuttal Fig. 3). This observation does not agree with the GAS FSTL1 overexpression data.
- 4) No studies are performed to understand how the FSTL1-DIP2A/CD14 pathway regulates lipogenic or fatty acid oxidation genes in the liver.
- 5) 1.25% cholesterol in the NASH diet remains a concern.

Reviewer #4 (Remarks to the Author):

General Comments

In the manuscript NCOMMS-22-47347A “Metabolic crosstalk between skeletal muscle cells and liver through IRF4-FSTL1 in nonalcoholic steatohepatitis”, the authors discuss a signaling pathway between IRF4-FSTL1-DIP2A/CD14. The focus of my review is specifically regarding the proteomic data.

Specific Comments

1. The methods section is inadequate and incomplete. Line 640 describes human serum method, but data all the proteomic data in Extended Figure 3 is from mouse serum and there is no mention of that in the methods. The citation to reference 59 is not sufficient. Please provide full details on how the sample was collected, prepared, and analyzed. For example, how much total protein was used (was that how the sample was normalized?). How was the serum collected and how was the protein extracted? Did you do an albumin removal of the serum? What were the digestion conditions (I assume proteins were digested into peptides, but there is no mention of that). Were protease inhibitor uses? What was the instrument parameters/and settings? How was the data searched an analyzed, what were the search parameters and what were the statistical analysis settings, did you use imputation and if so what kind? It is quite difficult to assess the quality of the data with no method details.
2. Regarding Extended Data Figure 3: The figure legend is not clear and there is no description for figures Ext 3f-h.

3. Please elaborate of Figure Ext 3b. What are the different colors? And what is Ext Fig 3c? All the text says is “screening of different expressed myokines”. I do not comprehend what that means and what the figure is.

The following are our point-by-point responses to the concerns and comments raised by the reviewers

REVIEWER COMMENTS

Reviewer #1 (Remarks to the Author):

I am pleased to say that the manuscript has significantly improved and has addressed my main concerns. Therefore, I fully support its publication without any further experiments. However, there are still a few issues that could be resolved by addressing the following points as caveats or limitations.

Responses: Thank you for the positive feedback on our revision. We appreciate your time and efforts on evaluating this manuscript. Please see below our responses to address your remaining concerns.

1. Reviewer three's point regarding the high cholesterol content (1.25%) in the NASH diet, compared to the typical cholesterol levels (0.21%) reported in the diet, is valid. As a result, it would be appropriate to include this as a limitation in the study and provide additional information on the macro- and micro-nutrient composition of the diet.

Responses: Thank you for the constructive suggestion for the diet issue, we have added the limitation of the study, especially about NASH diet. These additions are now described in the text (page 19, lines 489-490).

2. The authors used an fl/fl mouse as a control, but previous research (PMID: 29858650) has shown that activating Cre recombinase in skeletal muscle can improve glucose tolerance. Therefore, when investigating skeletal muscle gene/protein function in the context of obesity using Cre-LoxP systems, it's important to include a Cre mouse as a necessary control littermate. This should be taken into consideration when planning future studies.

Responses: Thank you for the constructive suggestion. In the previous study (PMID: 29858650), the authors used HSA-cre. In our study, we used Mly1-cre, both of which are muscle-specific cre, but are slightly different. We fully agree with the reviewer' comment, we will include a Cre mouse as a necessary control littermate in the future study.

I would like to extend my congratulations to the authors for producing such a well-crafted paper.

Responses: We highly appreciate Reviewer 1 for his/her time and efforts in reevaluating our manuscript, which help largely improve the quality of this manuscript!

Reviewer #2 (Remarks to the Author):

Authors addressed the questions raised by the reviewer.

Responses: We highly appreciate Reviewer 2 for his/her time and efforts in reevaluating our manuscript.

Reviewer #3 (Remarks to the Author):

Although the authors addressed some of the concerns, some other concerns remain in place:

Responses: We appreciate this reviewer's time and efforts on evaluating this manuscript. Please see below our responses to the remaining concerns.

1) Plasma FSTL1 levels in mice overexpressing IRF4 are not shown. The revised manuscript only presents plasma FSTL1 levels in mice lacking IRF4.

Responses: The plasma FSTL1 levels in mice from the IRF4 overexpression experiments are now provided. The results showed that the level of plasma FSTL1 was significantly elevated by IRF4 overexpression in mice fed on 24-weeks NASH diet.

Rebuttal Fig. 1

Elisa analysis of the level of plasma FSTL1 in WT, MOE mice (n=6). All results were shown as mean \pm SEM. *** p <0.001, compared with the WT group.

2) The new data (Rebuttal Fig. 17) show that the knockdown of DIP2A and CD14 receptors reduces liver steatosis and injury when FSTL1 is overexpressed in skeletal muscle, suggesting that FSTL1 regulates the development of NASH independent of the two receptors. Thus, how FSTL1 interacts with hepatocytes to regulate NASH development remains unclear.

Responses: We appreciate the Reviewer's insightful points. In our study, we found that ablation of IRF4 in skeletal muscle ameliorated NASH progression. However, when we introduced skeletal muscle-specific overexpression (AAV2/9-MCS-FSTL1 or AAV-MCK promoter-FSTL1) of FSTL1 in F4MKO mice, FSTL1 overexpression counteracted the effects on the liver induced upon the ablation of skeletal muscle IRF4. Then we used AAV delivery to directly block both FSTL1 receptor Dip2a and Cd14 expression, FSTL1 could not rescue hepatic damage, lipid accumulation, and fibrosis when both Dip2a and Cd14 were knocked down in the liver. Meanwhile, in vitro experiments, results showed that FSTL1 could not restore cellular lipid accumulation and lipid metabolism gene expression in Dip2a or Cd14 knockdown AML12 steatosis cells co-cultured with GAS from F4MKO mice. And following Cd14 knockdown in macrophages

and Dip2a knockdown in HSC-T6, rFSTL1 was unable to reverse inflammation-related gene expression in macrophages and fibrosis-related gene expression in HSC-T6 cells co-cultured with GAS from F4MKO mice. Moreover, the other receptors of FSTL1 did not get similar results in vitro experiments. In summary, these data revealed that FSTL1 regulates the development of NASH via the two receptors, suggesting that FSTL1 regulates the development of NASH is dependent of the two receptors.

3) AAV-shFSTL1 can cause the knockdown of FSTL1 in other cell types as well. It does not have a muscle-specific promoter to drive the FSTL1 knockdown. Importantly, the knockdown of GAS FSTL1 does not affect plasma ALT or AST levels or liver phenotype (Rebuttal Fig. 3). This observation does not agree with the GAS FSTL1 overexpression data.

Responses: We thank the reviewer for pointing this out. Because the most crucial question we would like to address was whether rescuing FSTL1 expression in IRF4MKO mice could abolish the ameliorated NASH effects, we have to focus on the effects of FSTL1 treatment in IRF4MKO mice. We used several models to prove that muscle derived IRF4 regulates NASH through FSTL1. First, overexpression of FSTL1 can abolish the ameliorated NASH phenotype caused by IRF4 muscle specific knockout; Second, overexpress FSTL in muscle aggravates NASH. FSTL1 knockdown in muscle did not alleviate the NASH phenotype probably for the following reasons: 1, the mRNA expression showed a similar trend over the same time period, although it did not achieve significant changes in liver function, perhaps extending the time would result in a consistent phenotype; 2, it is also possible that due to the presence of IRF4, knockdown of FSTL1 could not ameliorate NASH induced by NASH diet for the same length time; 3, we knockdown FSTL1 only in GAS but no other skeletal muscles, which may be not enough to see the effects in liver. We also added these possibilities to the limitations (page 13, lines 336-337). We are not sure if other effectors play roles in this progress. This may be a future direction for research.

4) No studies are performed to understand how the FSTL1-DIP2A/CD14 pathway regulates lipogenic or fatty acid oxidation genes in the liver.

Responses: Previous research (PMID: 33246164) reported that FSTL1/Dip2a activation could active its downstream AKT phosphorylation in cardiomyocyte. We also tested the level of AKT phosphorylation in our model. Results showed that there is no difference (Rebuttal Fig. 2), indicating that may be other effectors were involved. We also added these possibilities to the limitations (page 14, lines 337-339). As for how FSTL1-Dip2a/Cd14 regulates lipid metabolism in liver, it may be a good research direction in the future.

Rebuttal Fig. 2

Western blot analysis of the expression of p-AKT in liver of Flox, F4MKO mice, F4MKO+FSTL1,

F4MKO+FSTL1+DC KD mice on 24-weeks NASH diet.

5) 1.25% cholesterol in the NASH diet remains a concern.

Responses: As Reviewer one's suggestion, we added the high cholesterol in the NASH diet as one of limitation of this study. These additions are now described in the text (page 13, lines 334-336).

Reviewer #4 (Remarks to the Author):

General Comments

In the manuscript NCOMMS-22-47347A "Metabolic crosstalk between skeletal muscle cells and liver through IRF4-FSTL1 in nonalcoholic steatohepatitis", the authors discuss a signaling pathway between IRF4-FSTL1-DIP2A/CD14. The focus of my review is specifically regarding the proteomic data.

Responses: We thank Reviewer 4 for his/her time and efforts on evaluating this manuscript. Please see below our responses to the suggestions and comments.

Specific Comments

1. The methods section is inadequate and incomplete. Line 640 describes human serum method, but data all the proteomic data in Extended Figure 3 is from mouse serum and there is no mention of that in the methods. The citation to reference 59 is not sufficient. Please provide full details on how the sample was collected, prepared, and analyzed. For example, how much total protein was used (was that how the sample was normalized?). How was the serum collected and how was the protein extracted? Did you do an albumin removal of the serum? What were the digestion conditions (I assume proteins were digested into peptides, but there is no mention of that). Were protease inhibitor uses? What was the instrument parameters/and settings? How was the data searched an analyzed, what were the search parameters and what were the statistical analysis settings, did you use imputation and if so what kind? It is quite difficult to assess the quality of the data with no method details.

Responses: We are sorry for not providing enough experimental details during previous revision. During this revision, we have tried our best to add more details to all experiments in the Methods and/or in the Figure Legends. These methods details are now described in the text (page 25-28, lines 657-716).

2. Regarding Extended Data Figure 3: The figure legend is not clear and there is no description for figures Ext 3f-h.

Responses: We had updated the figure legend of Extended Data Figure 3 as Screening myokines responsible for IRF4-mediated NASH progression. We had described in detail what the Figure Ext 3a-e represents. In the new version of Extended Data Figure 3, there no existed Ext 3f-h. More details are now provided for figure legend of Extended Data Figure 3 (page 29, lines 750-759).

3. Please elaborate of Figure Ext 3b. What are the different colors? And what is Ext Fig 3c? All the text says is "screening of different expressed myokines". I do not comprehend what that means and what

the figure is.

Responses: We are sorry for not providing enough details for this Figure Ext 3. During this revision, we have added more details in the Figure Ext 3 Legends. In Figure Ext 3b, upregulated and downregulated proteins from proteomics were shown in red and green respectively. Upregulated and downregulated genes from RNAseq data were shown in bright blue and light blue respectively. Figure Ext 3c showed the overlap of different expressed proteins and myokines obtained from public literatures. Upregulated and downregulated proteins from proteomics were shown in red and blue respectively, and myokines obtained from public literatures were shown green (page 29, lines 750-756).

REVIEWERS' COMMENTS

Reviewer #3 (Remarks to the Author):

The authors addressed my points 1, 2, and 5. My points 3 and 4 are not addressed satisfactorily. However, this work may be accepted for publication.

Reviewer #4 (Remarks to the Author):

The responses to my review have strengthened the paper and I am satisfied with the changes.

The following are our point-by-point responses to the concerns and comments raised by the reviewers

REVIEWERS' COMMENTS

Reviewer #3 (Remarks to the Author):

The authors addressed my points 1, 2, and 5. My points 3 and 4 are not addressed satisfactorily. However, this work may be accepted for publication.

Responses: We highly appreciate Reviewer 3 for his/her time in evaluation of this study, which help largely improve the quality of this manuscript!

Reviewer #4 (Remarks to the Author):

The responses to my review have strengthened the paper and I am satisfied with the changes.

Responses: We greatly thank Reviewer 4 for his/her highly valuable comments and suggestions, which largely help on generating this significantly improved version of manuscript!

All the bests

Xingxing Kong

Professor